



# Measurement report: Evaluation of sources and mixing state of black carbon aerosol under the background of emission reduction in the North China Plain: implications for radiative effect

Qiyuan Wang[1,2], Li Li[1], Jiamao Zhou[1], Jianhuai Ye[3], Wenting Dai[1], Huikun Liu[1], Yong Zhang[1], Renjian Zhang[4,5], Jie Tian[1], Yang Chen[6], Yunfei Wu[4], Weikang Ran[1], and Junji Cao[1,2]

[1]Key Laboratory of Aerosol Chemistry and Physics, State Key Laboratory of Loess and Quaternary Geology, Institute of Earth Environment, Chinese Academy of Sciences, Xi'an 710061, China
[2]CAS Center for Excellence in Quaternary Science and Global Change, Xi'an 710061, China
[3]School of Engineering and Applied Sciences, Harvard University, Cambridge, MA 02138, USA
[4]Key Laboratory of Regional Climate-Environment Research for Temperate East Asia, Institute of Atmospheric Physics, Chinese Academy of Sciences, Beijing 100029, China
[5]Xianghe Observatory of Whole Atmosphere, Institute of Atmospheric Physics, Chinese Academy of Sciences, Xianghe County, Hebei Province, 065400, China
[6]Chongqing Institute of Green and Intelligent Technology, Chinese Academy of Sciences, Chongqing 400714, China

*Correspondence to*: Qiyuan Wang (wangqy@ieecas.cn) and Junji Cao (cao@loess.llqg.ac.cn)

**Abstract.** Accurate understanding of sources and mixing state of black carbon (BC) aerosol is essential for assessing its impacts on air quality and climatic effect. Here, a winter campaign (December 2017 – January 2018) was conducted in the North China Plain (NCP) to evaluate the sources, coating composition, and radiative effect of BC under the background of emission reduction since 2013. Results show that liquid fossil fuel source (i.e., traffic emission) and solid fuel source (i.e., biomass and coal burning) contributed 69% and 31% to the total BC mass, respectively, using a multiwavelength optical approach combined with the source-based aerosol absorption Ångström exponent values. The air quality model indicates that local emission was the dominant contributor to BC at the measurement site on average, however, emissions in the NCP exerted a critical role for high BC episode. Six classes of BC-containing particles were identified, including (1) BC coated by organic carbon and sulphate (52% of total BC-containing particles), (2) BC coated by Na and K (24%), (3) BC coated by K, sulphate, and nitrate (17%), (4) BC associated with biomass burning (6%), (5) Pure-BC (1%), and (6) others (1%). Different BC sources had distinct impacts on those BC-containing particles. A radiative transfer model estimated that





the amount of BC detected can produce an atmospheric forcing of +18.0 W m$^{-2}$ and a heating rate of 0.5 K day$^{-1}$. Results presented herein highlight that further reduction of solid fuel combustion-related BC may be a more effective way to mitigate regional warming in the NCP, although larger BC contribution was from liquid fossil fuel source.

## 1 Introduction

Over the past decades, black carbon (BC) aerosol has attracted considerable attention due to its substantial effects on climate and atmospheric environment (Bond et al., 2013). BC can affect climate via direct or semidirect radiative effects or indirect cloud-albedo effects (Boucher et al., 2013). Due to its strong light-absorbing ability, BC can produce a substantial climate forcing globally in the present-day atmosphere (+1.1 W m$^{-2}$), and hence it is considered as the second largest anthropogenic warming agent after carbon dioxide (Bond et al., 2013). Moreover, high atmospheric BC loading can depress the development of planetary boundary layer and aggravate haze pollution (Ding et al., 2016). Due to the short atmospheric residence time, reducing atmospheric BC loading is regarded as a win-win policy intervention to mitigate climate change and improves air quality (Kopp and Mauzerall, 2010).

Owing to BC's various emission sources (e.g., fossil fuel and biomass burning) and complex physicochemical properties (e.g., morphology, size, and coating composition), large uncertainty still remains in assessing its climate and environmental impacts (Vignati et al., 2010). For BC source apportionment, current methods are usually based on the data obtained from offline filter-based or online spectroscopy techniques (Briggs and Long, 2016). Among them, carbon isotope approach (e.g., $\Delta^{14}$C) and multi-wavelength optical method (e.g., aethalometer model) are often used to quantify the sources of BC (e.g., Zotter et al., 2017; Zhang et al., 2015). The carbon isotope approach could obtain relatively accurate result of BC source apportionment. However, the analysis is limited by time resolution of the filter samples. The aethalometer model which utilizes online data, has the advantage of superior time resolution in determining BC sources. The principle of the aethalometer model is based on the Beer-Lambert's Law using measured aerosol light absorption at different wavelengths (Sandradewi et al., 2008). However, due to the lack of source-specific aerosol absorption Ångström exponent (AAE), a number of studies often adopted AAEs in aethalometer model cited from previous literatures even the fuel types are





distinct among studies (e.g., Kalogridis et al., 2018; Zheng et al., 2019). This may induce a large uncertainty in the accuracy of BC source apportionment. Therefore, a diverse set of AAEs from different source emissions are needed to improve the performance of the aethalometer model.

As an important chemical property of aerosol, BC mixing state describes whether other chemical
composition is coated on BC particles (internally mixed) or exists as separate particles (externally mixed). Generally, freshly emitted BC particles (e.g., diesel vehicle emission) exhibit typical external mixing, but over time, they become gradually internally mixed with other non-BC materials (e.g., organics, sulphate, and nitrate) during atmospheric processes (Eriksson et al., 2017). Compared with uncoated BC particles, the coated ones can enhance the absorption efficiency of solar radiation by a factor of $1.2 - 2.0$, which is
strongly associated with the chemical composition of the coatings on BC particles and their thickness (Wang et al., 2014; Fierce et al., 2016; Liu et al., 2017). Although determining BC mixing state through observations is still challenging, emerging advances in online mass spectrometry technique make it possible for obtaining the chemical characteristics of BC coatings. Based on this method, some studies reported direct observations of chemical composition associated with BC particles as well as their
evolution features in the atmosphere (e.g., Zhang et al., 2014; Arndt et al., 2017).

As a hotspot region for anthropogenic BC emissions, China accounts up to 14% of the global BC radiative forcing (Li et al., 2016). Over the past decade, China has suffered from serious air pollution, especially in the North China Plain (NCP) (An et al., 2019). To improve air quality, the Chinese State Council has been promulgated a series of regulations to reduce air pollutants, e.g., the most rigorous regulation of the
'Action Plan for the Prevention and Control of Air Pollution' (APPCAP) released on 10 September 2013, which aims to reduce particulate matter by up to 10% by 2017 relative to 2012 levels for all prefecture-level cities in China. Previous studies have demonstrated the effectiveness of China's clean air actions in reduction of particulate matter (Zhang et al., 2019). The decreased BC and co-emitted pollutants can affect the interactions between BC and secondary aerosols which in turn results in changes in the
physicochemical properties of BC aerosol. In the context of emission reduction, assessment of physicochemical characteristics of BC will be useful for improving our understanding of anthropogenic climate impacts in current China. However, studies focused on this aspect are limited at present. Therefore, we conducted intensive measurements during winter in the last year of the APPCAP at a regional site in



the NCP region to (1) determine the contributions of different sources and regions to BC mass, (2) identify the chemical composition of BC coatings, and (3) evaluate the impacts of BC on radiative effect.

## 2 Methodology

### 2.1 Sampling site

The Xianghe Atmospheric Integrated Observatory (39°45' N, 116°57' E) is located in a small county of Xianghe, the north of the NCP (Fig. S1). The county has an area of 458 km$^2$ and a total population of 0.36 million. It is bordered by Beijing in the northwest (~45 km) and Tianjin in the southeast (~79 km). This place is considered to be a regional observation site which suffers from frequent pollution plumes (e.g., urban, rural, or mixed origins) from surrounding areas (Wang et al., 2019a). Intensive measurements were

conducted in winter from 1 December 2017 to 31 January 2018. During the campaign, the weather was cold (temperature = -1°C) and dry (relative humidity = 35%).

### 2.2 Field Observations

### 2.2.1 Optical measurements

A model AE33 aethalometer (Magee Scientific, Berkeley, CA, USA) was used to measure the aerosol

light absorption coefficients at multi-wavelengths ($b_{abs}(\lambda)$) with a PM$_{2.5}$ cyclone (SCC 1.829, BGI Inc. USA). Drinovec et al. (2015) give a detailed description of the operation principle of this instrument. Briefly, seven different light sources were utilized to irradiate the filter deposition spot, and the light attenuation is detected by optical detectors. Although the biases caused by nonlinear loading effect have been resolved by AE33 aethalometer, the impact of filter matrix scattering can still interfere the

measurement accuracy (Drinovec et al., 2015). Thus, a photoacoustic extinctiometer (PAX) operating at $\lambda$ = 532 nm was installed in parallel with the AE33 aethalometer to correct this artifact. More detailed operating principle and calibrating procedure of PAX are described in our previous publication (Wang et al., 2018a). As shown in Fig. S2, a similar wavelength (520 nm) of AE33 absorption was correlated strongly with the PAX absorption (R$^2$ = 0.97, $p$ < 0.01), and the slope of 2.6 was then used to correct the

AE33 data.

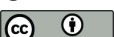



### 2.2.2 Measurement of BC mixing state

The chemical composition of BC coatings was determined by a real-time single particle aerosol mass spectrometer (SPAMS, Hexin Analytical Instrument Co., Ltd., Guangzhou, China). The ambient aerosol is drawn into the evacuated system via a critical orifice (100 μm) with a flowrate of 0.08 L min$^{-1}$; after

passing through an aerodynamic lens, the sampled particles are accelerated to certain speeds. Two diode Nd:YAG lasers (MLL-III-532, Changchun, China) operated at 532 nm are used to determine the aerodynamic diameters (0.2 – 2.0 μm) of passing particles. After then, a pulsed 266 nm Nd:YAG laser (UL728F11-F115, Quantel, France) is used to ionize those particles. Finally, a dual-polarity time-of-flight mass spectrometer is applied to detect those generated positive and negative fragment ions. A MATLAB-

based YAADA toolkit (www.yaada.org) was used to search and analyze the imported single-particle mass spectral data. An adaptive resonance theory-based neural network algorithm (ART-2a) was utilized to perform particle clustering, which set a vigilance factor of 0.8, a learning rate of 0.05, and 20 iterations (Li et al., 2019).

### 2.2.3 Measurement of organic aerosol

The mass concentrations of organic aerosol (OA) in submicron particles were obtained with a aerosol chemical speciation monitor (ACSM, Aerodyne Research Inc., Billerica, MA, USA). Ambient particles were drawn into the ACSM system through a Nafion$^®$ dryer (MD-700-24S-1; Perma Pure, Inc.) to avoid the influence of particle collection efficiency that caused by water condensation in the sampling line. The principle of ACSM has been elaborated elsewhere (Ng et al., 2011). In brief, the sampled particles are

focused into a beam through a critical orifice of 100 μm diameter; then those non-refractory particles are vaporized on a hot surface (~600 °C) and ionized with 70 eV electrons; and finally, the mass fragments are detected by a quadrupole mass spectrometer. The primary OA was further resolved into biomass burning OA (BBOA), coal combustion OA (CCOA), and hydrocarbon-like OA (HOA). Detailed information regarding OA source apportionment can be found in supplemental material of Wang et al.

(2019b).





## 2.3 Source emission experiments

A custom-made passivated aluminum chamber (~8 m$^3$) was used to characterize the emissions of solid fuels (i.e., biomass and coal). The emitted smokes were diluted by a Model 18 dilution sampler (Baldwin Environmental Inc., Reno, NV, USA) before AE33 aethalometer measurements. The performance evaluation of this chamber is given in Tian et al. (2015). Several residues of alimentary crops (e.g., wheat straw, rice straw, and corn stalk), economic crops (e.g., cotton stalk, sesame stalk, and soybean straw), and firewood were used to represent the biomass burning occurred in the NCP. Each weighted sample was burned on a platform that placed inside the combustion chamber (Wang et al., 2018b). Meanwhile, bituminous coal and honeycomb briquet that commonly used in the NCP were collected in Shanxi and Shaanxi Provinces. A typical stove that extensively used in the NCP was applied for coal test burns inside the combustion chamber (Tian et al., 2019). Furthermore, motor vehicle exhaust emissions were conducted using bench tests. Gasoline and diesel cars at idle speed and several different driving speeds (i.e., 20 and 40 km h$^{-1}$) were tested.

## 2.4 Data analysis methods

### 2.4.1 Aethalometer model

An aethalometer model proposed by Sandradewi et al. (2008) was applied to quantify the contributions of liquid fossil fuels (i.e., gasoline and diesel for traffic emission) and solid fuels (i.e., biomass and coal) to BC mass at Xianghe. The measured $b_{abs}(\lambda)$ at the wavelengths of 370 nm ($b_{abs}(370)$) and 880 nm ($b_{abs}(880)$) were used in the model. As demonstrated previously, $b_{abs}(\lambda)$ can be contributed by carbonaceous aerosols and mineral dust. Due to the small mass fraction of mineral dust in PM$_{2.5}$ during the campaign (11% of PM$_{2.5}$ mass) and its small mass absorption cross section (0.09 m$^2$ g$^{-1}$ at $\lambda$ = 370 nm and 0.001 m$^2$ g$^{-1}$ at $\lambda$ = 880 nm, Yang et al., 2009), the light absorption caused by mineral dust can be neglected. Previous studies have demonstrated that the $b_{abs}(880)$ is mainly contributed by BC aerosol, while $b_{abs}(370)$ is associated with BC, primary and secondary brown carbon (pBrC and sBrC, respectively) (Laskin et al., 2015). Finally, the $b_{abs}(370)$ and $b_{abs}(880)$ can be calculated from the perspective of emission sources as follows:





$$b_{abs}(880)_{lff} + b_{abs}(880)_{sf} = b_{abs}(880) \tag{1}$$

$$b_{abs}(370)_{lff} + b_{abs}(370)_{sf} = b_{abs}(370) - b_{abs}(370)_{sBrC} \tag{2}$$

$$\frac{b_{abs}(370)_{lff}}{b_{abs}(880)_{lff}} = \left(\frac{370}{880}\right)^{-AAE_{lff}} \tag{3}$$

$$\frac{b_{abs}(370)_{sf}}{b_{abs}(880)_{sf}} = \left(\frac{370}{880}\right)^{-AAE_{sf}} \tag{4}$$

where $b_{abs}(880)_{lff}$ and $b_{abs}(880)_{sf}$ refer to the light absorption of BC at $\lambda = 880$ nm derived from liquid fossil fuel and solid fuel sources, respectively; $b_{abs}(370)_{lff}$ and $b_{abs}(370)_{sf}$ represent the light absorption of BC and pBrC at $\lambda = 370$ nm emitted from liquid fossil fuel and solid fuel sources, respectively; $b_{abs}(370)_{sBrC}$ is representative of the BrC absorption caused by secondary formation processes, which was estimated by a BC-tracer method combined with a minimum $R$-squared approach (Wang et al., 2019b);

and $AAE_{lff}$ and $AAE_{sf}$ describe the aerosol AAEs from liquid fossil fuel and solid fuel sources, respectively.

After obtaining $b_{abs}(880)_{lff}$ and $b_{abs}(880)_{sf}$, the mass concentrations of BC contributed by liquid fossil fuel and solid fuel sources ($BC_{lff}$ and $BC_{sf}$, respectively) were then calculated by dividing the mass absorption cross section of BC at $\lambda = 880$ nm ($MAC_{BC}(880)$):

$$BC_{lff} = \frac{b_{abs}(880)_{lff}}{MAC_{BC}(880)} \tag{5}$$

$$BC_{sf} = \frac{b_{abs}(880)_{sf}}{MAC_{BC}(880)} \tag{6}$$

The underlying assumption of Eqs. (5) and (6) was that $MAC_{BC}(880)$ was the same as those from liquid fossil fuel and solid fuel sources, which is justified by Zotter et al. (2017). Due to a strong correlation between $b_{abs}(880)$ and EC mass concentration ($R^2 = 0.86$, $p < 0.01$, Fig. S3), the daily $MAC_{BC}(880)$ can

be estimated by daily $b_{abs}(880)$ dividing by the corresponding EC mass concentration.

### 2.4.2 Regional chemical dynamical model

The local versus regional contributions to BC mass at Xianghe was quantified by the Weather Research and Forecasting model coupled with Chemistry (WRF-Chem). Here BC used as a tracer was added into the WRF-Chem to improve model's operation efficiency (Zhao et al., 2015). More detailed descriptions





regarding the model configurations are shown in Text S1. The performance of WRF-Chem simulation was evaluated with a mathematical parameter of index of agreement (IOA), which describes the relative difference between the simulated and observed values. The IOA can vary from 0 to 1, with the value closer to 1 meaning the better performance of the model simulation. The IOA is calculated as follows (Li

et al., 2011):

$$IOA = 1 - \frac{\sum_{i=1}^{N}(S_i - O_i)^2}{\sum_{i=1}^{N}(|S_i - S_{ave}| + |O_i - O_{ave}|)^2} \tag{7}$$

where $S_i$ and $O_i$ are the simulated and observed BC loadings, respectively; $S_{ave}$ and $O_{ave}$ represent the average value of simulated and observed BC loadings, respectively; and $N$ denotes the number of simulations.

**2.4.3 Estimations of radiative forcing and heating rate**

A Santa Barbara DISORT Atmospheric Radiative Transfer (SBDART) model that developed by Ricchiazzi et al. (1998) was used to perform the radiative transfer calculations in the shortwave spectral region (0.25 – 4.0 μm). The SBDART model is a widely used tool for estimating the aerosol direct radiative forcing (DRF) by the atmospheric science community (e.g., Raju et al., 2016; Singh et al., 2016;

Zhang et al., 2017; Rajesh and Ramachandran, 2018; Boiyo et al., 2019). The detailed principle of SBDART model can be found in Ricchiazzi et al. (1998). The DRFs induced by BC alone (or total aerosol) at the Earth's surface (ES), the top of the atmosphere (TOA), and in the atmosphere were estimated by the difference in the net flux with and without BC (or aerosol) under cloud-free conditions.

The optical properties of aerosol optical depth (AOD), single scattering albedo (SSA), and asymmetric

parameter (ASP) are essential input parameters in the SBDART model, and they were determined by an Optical Properties of Aerosol and Cloud (OPAC) model (Hess et al., 1998). The measured BC and water-soluble inorganic ions as well as the estimated water-soluble organic matter (assuming that the water-soluble organic matter accounts for 49% of organic matter based on the result of Zhang et al., 2018) and water insoluble matter (calculated by the mass concertation of PM$_{2.5}$ minus BC and water-soluble matter)

were used as input parameters in OPAC to reconstruct the AOD, SSA, and ASP. The difference between the reconstructed SSA by OPAC and measured SSA by PAX was 1.4%, indicating a reasonable reconstruction result.





Further, the atmospheric heating rate ($\frac{\partial T}{\partial t}$, in unit of K d$^{-1}$) induced by BC was calculated using the first law of thermodynamics and hydrostatic equilibrium as follows (Liou, 2002):

$$\frac{\partial T}{\partial t} = \frac{g}{C_p} \times \frac{\Delta F}{\Delta P} \tag{8}$$

where $\frac{g}{C_p}$ describes the lapse rate, of which g is the acceleration due to gravity and $C_p$ represents the specific heat capacity of air at a constant pressure (1006 J kg$^{-1}$ K$^{-1}$); $\Delta F$ is the atmospheric forcing induced by BC aerosol; and $\Delta P$ is representative of the atmospheric pressure difference, which was assumed to be 300 hPa.

## 3 Results and discussion

### 3.1 BC source apportionment

#### 3.1.1 Determination of source-specific AAEs

Table 1 shows the characteristics of AAEs obtained from source experiments of liquid fossil fuels and solid fuels. The average AAE$_{lff}$ was 1.3 ± 0.2, with higher values for gasoline car (1.4 – 1.5) than those for diesel engine car (1.1 – 1.2). Compared with AAE$_{lff}$, larger average value was found for AAE$_{sf}$ (2.8 ± 1.0) with the largest one obtained from honeycomb briquet emissions (4.0 ± 0.9), followed by firewood burning (2.9 ± 0.2), crop residues emissions (2.4 ± 0.4), and bituminous coal emissions (1.1 ± 0.2). The large variabilities in AAE$_{sf}$ were potentially affected by the various types of solid fuels and their burning conditions; for example, AAE$_{sf}$ shows a weak but significant inverse correlation with combustion efficiency at 95% confidence interval ($R^2$ = 0.14, $p < 0.05$, Fig. S4).

The obtained average AAE$_{lff}$ (1.3) and AAE$_{sf}$ (2.8) were applied in the aethalometer model to obtain the BC source apportionment. The OA subtypes contributed by liquid fossil fuel (i.e., HOA) and solid fuel sources (i.e., BBOA + CCOA) were used to verify the reliability of model results. In the light of the range of source-based AAEs, a series of AAE$_{lff}$ and AAE$_{sf}$ were put into the aethalometer model to obtain the mass concentrations of BC$_{lff}$ and BC$_{sf}$. The correlations were then established for BC$_{lff}$ versus HOA and BC$_{sf}$ versus BBOA + CCOA (Fig. 1). The variations in AAE$_{lff}$ cannot affect the correlation between BC$_{lff}$ and HOA at a fixed AAE$_{sf}$, but their $R^2$ increased as AAE$_{sf}$ increased before 3; after the AAE$_{sf}$ larger than





3, the $R^2$ kept constant regardless of the variability in $AAE_{sf}$. In contrast, at a fixed $AAE_{lff}$, the $R^2$ between $BC_{sf}$ and BBOA + CCOA was independent of the $AAE_{sf}$ variation. For the used $AAE_{lff}$ (1.3) and $AAE_{sf}$ (2.8), the coefficient of determination of $BC_{lff}$ versus HOA ($R^2 = 0.60$) and $BC_{sf}$ versus BBOA + CCOA ($R^2 = 0.66$) belonged to the upper limit of all the $R^2$ values obtained from different ranges of $AAE_{lff}$ and

$AAE_{sf}$ (Fig. 1). Based on the BC source apportionment result, the estimated ratios of HOA/$BC_{lff}$ (1.7) and (BBOA + CCOA)/$BC_{sf}$ (8.4) are comparable to those calculated with emission factors (Cheng et al., 2010; Sun et al., 2018). Therefore, the pair of $AAE_{lff}$ of 1.3 and $AAE_{sf}$ of 2.8 is a reasonable selection for this study.

### 3.1.2 Characteristics of $BC_{lff}$ and $BC_{sf}$

The average BC mass concentration was $3.6 \pm 4.0$ µg m$^{-3}$ and varied largely from 0.1 to 24.4 µg m$^{-3}$ during the campaign (Fig. 2a). The estimated $BC_{lff}$ comprised 69% (2.5 µg m$^{-3}$) of BC loading, which was over two times larger than the contribution of $BC_{sf}$ (31%, 1.1 µg m$^{-3}$) (Fig. 2b). This indicates that traffic emissions were the dominant contributor to BC mass at Xianghe. As shown in Fig. 2c, the diurnal variation of $BC_{lff}$ exhibited two peaks at 08:00 and 19:00, which were coincided with the morning and

evening rush-hour traffic. Although $BC_{sf}$ also showed two peaks appeared in the same period with $BC_{lff}$, they were affected by the residential cooking activities in surrounding rural areas, where solid fuel is a commonly used household energy (Liu et al., 2016). Both decreased $BC_{lff}$ and $BC_{sf}$ were observed in the afternoon owing to the increases of planetary boundary layer height and wind speed (Fig. 2d, data sources see Text S2). In contrast to a rapid decline in $BC_{lff}$ after 19:00, $BC_{sf}$ remained at a high level until midnight.

Meanwhile, the $BC_{sf}$/BC fraction was also increased towards to the night, indicating the enhanced heating activities with the use of solid fuel on cold nights.

Fig. 3 shows some previous studies regarding the BC source apportionment, and their detailed information is summarized in Table S1. Liquid fossil fuel source was an absolute major contributor to BC mass at urban area due to the heavy traffic, but the contribution of solid fuel source enhances at rural area, where

wood burning is commonly used as a household energy. Furthermore, we can also find that the dominant BC source in winter NCP has been changed from past solid fuel to current liquid fossil fuel, even though uncertainty may be caused by the limited studies in the NCP. This change is probably attributed to the



rigorous regulations promulgated by the Chinese State Council, of which the large-scale project of coal-to-gas switching has been considered as an effective way to reduce pollutants. In addition, although high-emission motor vehicles are also banned, the total number of vehicles in the NCP region increase largely from 38.7 million in 2013 to 60.3 million in 2017 (NBS, 2018). Therefore, the liquid fossil fuel source has become a more important contributor to BC compared with the solid fuel source.

## 3.2 Contribution of regional transport to BC

The period of 2 – 23 January 2018 was arbitrary selected to explore the regional contributions to BC loading at Xianghe using the WRF-Chem model. As shown in Fig. S5, the simulated BC mass concertation correlated significantly with the measured value ($R^2 = 0.61$, $p < 0.01$), and the IOA was estimated to be 0.72, indicating that BC formation process was succeed in catching by WRF-Chem. To determine the impacts of local emission and regional transport on BC, we set up six possible BC source regions as part of the modelling exercise (Fig. S6), and their location information is summarized in Table S2. The percent contributions from local emission and regional transport to BC mass are shown in Fig. 4. Although the contributions from different regions varied from day to day, the average BC mass contribution from local emission (53%) was slightly higher than that from regional transport (47%), including 20% from Beijing, 5% from Tianjin, 11% from NCP, 9% from northern Hebei, and 2% from other regions.

Although the local emission was the largest contributor to BC mass on average for all the simulated days, its contribution exhibited a negative correlation with the BC loading (Fig. 5a), indicating an increasing importance of regional transport when the high BC episode was occurred. Actually, only BC contributions from Tianjin and NCP increased as the BC loading increased (Fig. 5 c and d), suggesting that the south of Xianghe was an important pollution region to large BC mass at the sampling site. Taking the highest BC loading episode of 12 – 13 January as an example to explain the regional transport (Fig. 6). On 11 January before the high BC loading episode, strong north-westerly winds prevailed in the north of Xianghe, and about 44% and 32% of the total BC mass were contributed by local emission and Beijing as represented by Region 1 and 2, respectively. Thereafter, on 12 January, the winds turned to the southwest and passed over the NCP region. The BC loadings increased sharply, with regional transport





accounting for 83% of the total BC mass, of which the NCP region accounted for 63% (Fig. 4). On 13 January, the winds switched to south over the NCP region but decreased near the sampling site. The contribution of regional transport reduced to 66% of the total BC mass with 40% from NCP region and 15% from Tianjin (Fig. 4). On 14 January after the high BC episode, the winds turned to the northwest

and then the mass concentration of BC decreased gradually.

## 3.3 Chemical composition of BC coatings

The chemical composition of BC coatings was determined with a SPAMS. A total of 454433 particles whose mass spectra had obvious BC fragment ions (e.g., $m/z$ ±12, ±24, ±36, ±48, ±60, and so on) were identified as BC-containing particles. Further, six categories including BC-OCSOx, BC-NaK, BC-

KSOxNOx, BC-BB, Pure-BC, and BC-others were classified based on their mass spectral features. The average mass spectral pattern of each class is shown in Fig. 7, and the contribution of each class to the total BC-containing particles is summarized in Table 2.

The BC-OCSOx was characterized by obvious organic carbon (OC) signals in the positive mass spectrum (for example, intense signals of $^{37}C_3H^+$, $^{39}C_3H_3^+$, and $^{50}C_4H_2^+$ as well as moderate signals of $^{27}C_2H_3^+$,

$^{51}C_4H_3^+$, and $^{63}(CH_3)_2NH_2OH^+$) and strong sulphate ($^{97}HSO_4^-$) signal in the negative mass spectrum. This group was the largest contributor, constituting 52% of the total BC-containing particles (Table 2), indicating that BC was mainly coated with OC and sulphate. The presences of $^{43}C_2H_3O^+$ (a marker denoting the oxidized organics, Gunsch et al., 2018) and $^{97}HSO_4^-$ imply that BC-OCSOx was underwent a certain degree of atmospheric aging processes. As shown in Fig. 8, the BC-OCSOx number fraction

increased as $BC_{sf}$ increased. In contrast, it dropped when $BC_{lff}$ larger than the value of 75th percentile of $BC_{lff}$. This indicates a greater impact of solid fuel source on BC-OCSOx at a high BC loading environment compared with the liquid fossil fuel source. The diurnal variation in BC-OCSOx number fraction exhibited an upward trend at night after 19:00 (Fig. 9), which was attributed to the intensive domestic heating activities in surrounding rural areas.

The BC-NaK exhibited strong signals of $^{23}Na^+$ and $^{39}K^+$ in the positive mass spectrum and less intense signals of $^{26}CN^-$, $^{46}NO_2^-$, $^{62}NO_3^-$, and $^{97}HSO_4^-$ in the negative mass spectrum. This group was the second largest contributor accounting for 24% of total BC-containing particles (Table 2). Intense signals of BC



fragment ions (*m/z* 24, 36, 48, 60, and 72) were concentrated on the negative mass spectrum, indicating that BC-NaK particles were mainly freshly emitted. Meanwhile, relatively higher signal was found for nitrate than sulphate in the negative mass spectrum, although their signals were low. This is consistent with the motor vehicle emission, which contains substantial nitrogen oxides (May et al., 2014).

Furthermore, the number fraction of BC-NaK enhanced as $BC_{lff}$ increased but kept stable with $BC_{sf}$ (Fig. 8). These results demonstrate that BC-NaK was more likely associated with the fresh traffic emission relative to the solid fuel emission.

The BC-KSOxNOx had a strong signal of $^{39}K^+$ in the positive mass spectrum and intense signals of $^{46}NO_2^-$, $^{62}NO_3^-$, and $^{97}HSO_4^-$ in the negative mass spectrum. This group comprised 17% of total BC-containing

particles (Table 2). The high signal intensities of nitrate and sulphate indicate that BC-KSOxNOx particles suffered substantial aging processes in the atmosphere. The number fraction of BC-KSOxNOx was the only class that increased in the afternoon, which was consistent with the increase of ozone ($O_3$, Fig. 9) measured with an ultraviolet photometric Model 49*i* $O_3$ analyzer (Thermo Fisher Scientific, San Jose, CA, USA). This indicates that BC particles were easier to coated with sulphate and nitrate in the more

oxidation environment. The strong $^{39}K^+$ ion signal in the BC-KSOxNOx particles may imply a partial influence of biomass-burning emissions (Bi et al., 2011).

BC-BB contributed only 6% to total BC-containing particles (Table 2), and it was characterized by signals of $^{39/41}K^+$ in the positive mass spectrum and $^{26}CN^-$, $^{46}NO_2^-$, and $^{97}HSO_4^-$ in the negative mass spectrum. Several levoglucosan signals of $^{45}CHO_2^-$, $^{59}C_2H_3O_2^-$, and $^{73}C_3H_5O_2^-$ were also existed in the negative mass

spectrum, suggesting a typical biomass-burning characterization. Intense signals of negative ion spectrum of BC (e.g., *m/z* -24, -36, and -48) and relatively low signals of nitrate and sulphate indicate that BC-BB particles were less aged in the atmosphere. Due to the intensive heating activities, this class had an increased contribution to total BC-containing particles at night (Fig. 9).

The Pure-BC was mainly characterized by BC fragment ions (e.g., *m/z* ±24, ±36, ±48, ±60, and ±72).

Low ion signals of nitrate and sulphate indicate that Pure-BC was freshly emitted. This group contributed minor (1%, Table 2) to the total BC-containing particles. The BC-others was characterized by some metallic signals (e.g., $^{40}Ca^+$, $^{56}Fe^+/CaO^+$, and $^{62}FeO^+$) and aromatic signatures (e.g., $^{51}C_4H_3^+$, $^{63}C_5H_3^+$, $^{77}C_6H_5^+$, and $^{91}C_7H_7^+$) in the positive mass spectrum and strong signal of $NO_2^-$ (*m/z* 46) in the negative



mass spectrum. This indicates that this type of BC particles experienced a certain degree of aging processes in the atmosphere and internally mixed with metals and aromatic compounds. This class accounted for only 1% of total BC-containing particles (Table 2).

## 3.4 Implication for radiative effect

The variation in BC forcing effect was highly associated with BC burden in each day. As shown in Fig. 10, due to reduction in radiative energy reaching the surface through absorbing incoming sunlight, BC had a cooling effect of $-13.6 \pm 7.0$ W m$^{-2}$ at the ES, ranging from -1.9 to -27.9 W m$^{-2}$. In contrast, BC had a warm effect of $+4.4 \pm 3.0$ W m$^{-2}$ at the TOA, varying from +0.6 to +20.8 W m$^{-2}$, indicating a net energy gain. This was partly attributed to the strong BC absorption that can impede the back scattered radiation reaching the TOA.

The difference between BC DRF at the TOA and the ES gave the atmospheric forcing (a net atmospheric absorption) of $+18.0 \pm 9.6$ W m$^{-2}$, which corresponded to a heating rate of 0.5 K day$^{-1}$. The BC DRF in the atmosphere accounted for 86% of total aerosol DRF (+21.0 W m$^{-2}$), suggesting that BC has significant impact on perturbing the Earth-atmosphere radiative balance. The atmospheric heating in conjunction with the surface reduction in solar flux may aggravate the low-level inversion, leading to slowdown of thermal convection and in turn reducing the process of cloud formation (Chou et al., 2002).

As shown in Fig. 10c and d, the mean BC DRF caused by liquid fossil fuel (solid fuel) source was -7.0 W m$^{-2}$ (-5.4 W m$^{-2}$) at the ES and +2.1 W m$^{-2}$ (+1.7 W m$^{-2}$) at the TOA, producing an atmospheric forcing of +9.1 W m$^{-2}$ (+7.1 W m$^{-2}$). Due to stronger BC DRF, the average heating rate of the atmosphere caused by liquid fossil fuel source (0.3 K day$^{-1}$) was 33% more than solid fuel source (0.2 K day$^{-1}$). Although larger BC DRF was found for liquid fossil fuel source, its atmospheric forcing generated per unit BC mass concentration (3.6 (W m$^{-2}$) ($\mu$g m$^{-3}$)$^{-1}$) was 81% smaller compared with the solid fuel source (6.5 (W m$^{-2}$) ($\mu$g m$^{-3}$)$^{-1}$). This implies that reduction of solid fuel BC may be a more effective way to mitigate regional warming in the NCP, though BC emission from motor vehicles is also needed to be controlled.



## 4 Conclusions

The sources, coating composition, and radiative effect of BC were investigated during winter in the last year of the APPCAP at a regional site in the NCP. Based on the source-specific AAEs ($AAE_{lff} = 1.3$ and $AAE_{sf} = 2.8$), about 69% of BC was contributed by liquid fossil fuel source while the rest 31% was

produced by solid fuel source using an aethalometer model. The $BC_{lff}$ and $BC_{sf}$ both showed two peaks at 08:00 and 19:00 but with different causes. Due to rigorous regulations, the dominant BC source in winter NCP maybe changed from past solid fuels to current liquid fossil fuels. The WRF-Chem model shows that local emission (53%) was the largest contributor to BC loading on average, followed by Beijing (20%), Tianjin (5%), NCP (11%), northern Hebei (9%), and other regions (2%).

Based on the mass spectral characteristics, BC coated with OC and sulphate (BC-OCSOx) was the largest contributor (52%) to the measured BC-containing particles. The presences of $^{43}C_2H_3O^+$ and $^{97}HSO_4^-$ imply that BC-OCSOx was underwent a certain degree of atmospheric aging process. The solid fuel source had a greater impact on BC-OCSOx at a high BC loading environment compared with the liquid fossil fuel source. BC coated with Na and K (BC-NaK) was the second largest contributor to total BC-

containing particles (24%), and this class was more likely associated with the fresh traffic emissions relative to the solid fuel emissions. BC coated with K, sulphate, and nitrate (BC-KSOxNOx) accounted for 17% of the total BC-containing particles. This class was partially influenced by biomass-burning emissions and suffered substantial aging process in the atmosphere. The rest classes of BC-BB, Pure-BC, and BC-other contributed minor to total BC-containing particles (1 – 6%).

The SBDART model shows that BC can induce a cooling effect of -13.6 W m$^{-2}$ at the ES and a warming effect of +4.4 W m$^{-2}$ at the TOA. The difference between BC DRF at the TOA and ES gave the atmospheric forcing of +18.0 ± 9.6 W m$^{-2}$, which can produce a heating rate of 0.5 K day$^{-1}$. The atmospheric forcing of BC contributed 86% to the forcing caused by total aerosol in the atmosphere, suggesting that BC had a significant impact on perturbing the Earth-atmosphere radiative balance. From

the perspective of BC sources, the atmospheric forcing of BC was higher for liquid fossil fuel source than the solid fuel source, but the atmospheric forcing generated per unit BC mass concentration was larger for solid fuel source. This indicates that reduction of solid fuel BC may be a more effective way to further mitigate the regional warming in the NCP compared with BC that emitted from motor vehicles.



*Data availability.* The data presented in this study are available at the Zenodo data archive https://doi.org/10.5281/zenodo.3923612.

*Supplement.* The supplement related to this article is available online.

*Author contributions.* QW and JC designed the campaign. QW and WR carried out the field measurements. YZ and WR conducted the source experiments. QW, LL, JZ, and JY analyzed the data. QW drafted the paper. All the authors reviewed and commented on the paper.

*Competing interests.* The authors declare that they have no conflict of interest.

*Acknowledgments.* This study was supported by the Strategic Priority Research Program of Chinese Academy of Sciences (XDB40030200), the West Light Foundation of the Chinese Academy of Sciences (XAB2018B03), and the Youth Innovation Promotion Association of the Chinese Academy of Sciences (2019402).

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





**Table 1.** Summary of aerosol absorption Ångström exponent (AAE) obtained from source emission experiments.

| Type | Sub-type | Name | AAE | S.D.[a] | Test number |
|---|---|---|---|---|---|
| Solid fuel | Biomass | wheat | 2.7 | 0.3 | 5 |
| | | corn | 2.9 | 0.4 | 5 |
| | | rice | 2.5 | 0.2 | 6 |
| | | soybean | 2.1 | 0.1 | 4 |
| | | cotton | 2.4 | 0.3 | 4 |
| | | sesame | 2.3 | 0.3 | 4 |
| | | sugarcane | 1.6 | 0.02 | 2 |
| | | wood | 2.9 | 0.2 | 4 |
| | Coal | bituminous coal | 1.1 | 0.2 | 4 |
| | | honeycomb briquet | 4.0 | 0.9 | 16 |
| Liquid fuel | traffic | gasoline | 1.5 | 0.1 | 3 |
| | | diesel | 1.2 | 0.1 | 7 |

[a]S.D. represents standard deviation.



**Table 2.** Summary of names, numbers, and fractions of six types of black carbon (BC)-containing particles determined by a single particle aerosol mass spectrometer.

| Group | Number of particles | Fraction of particles (%) |
|---|---|---|
| BC internally mixed with OC and sulphate (BC-OCSOx) | 235874 | 51.9 |
| BC internally mixed with Na and K (BC-NaK) | 107272 | 23.6 |
| BC internally mixed with K, sulphate, and nitrate (BC-KSOxNOx) | 75227 | 16.6 |
| BC from biomass burning (BC-BB) | 26307 | 5.8 |
| Pure-BC | 5083 | 1.1 |
| Unidentified BC (BC-others) | 4670 | 1 |
| Total BC-containing | 454433 | 100 |





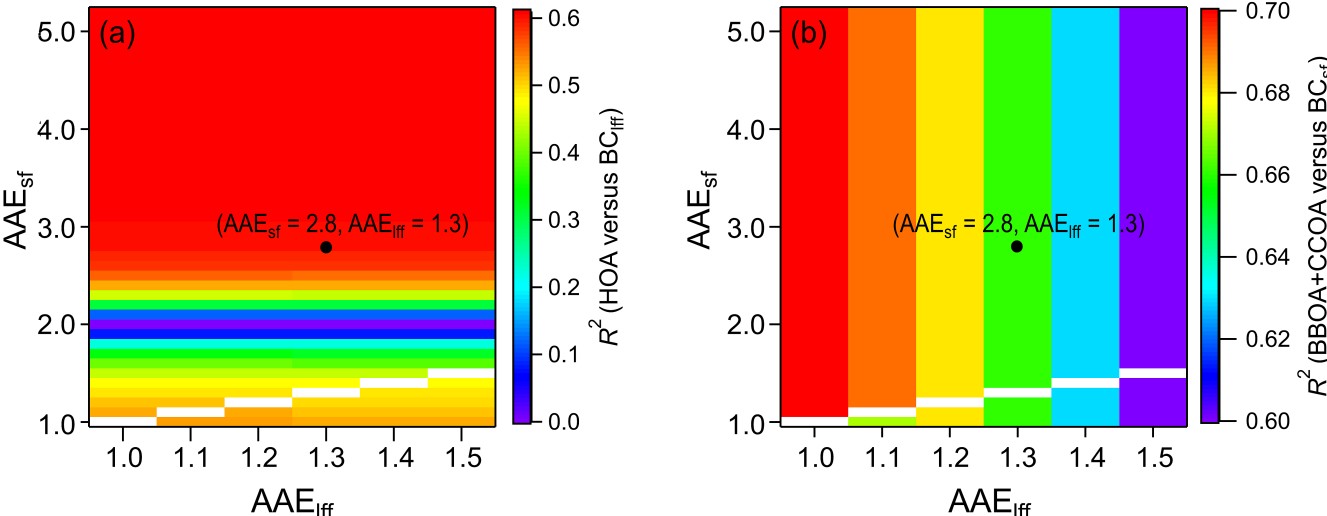

**Figure 1.** Distributions of the coefficient of determination ($R^2$) of (a) hydrocarbon-like organic aerosol (HOA) versus black carbon (BC) from liquid fossil fuel source ($BC_{lff}$) and (b) sum of the mass concentrations of biomass burning and coal combustion organic aerosols (BBOA + CCOA) versus BC from solid fuel source ($BC_{sf}$) at different absorption Ångström exponent of liquid fossil fuel ($AAE_{lff}$) and solid fuel sources ($AAE_{sf}$).





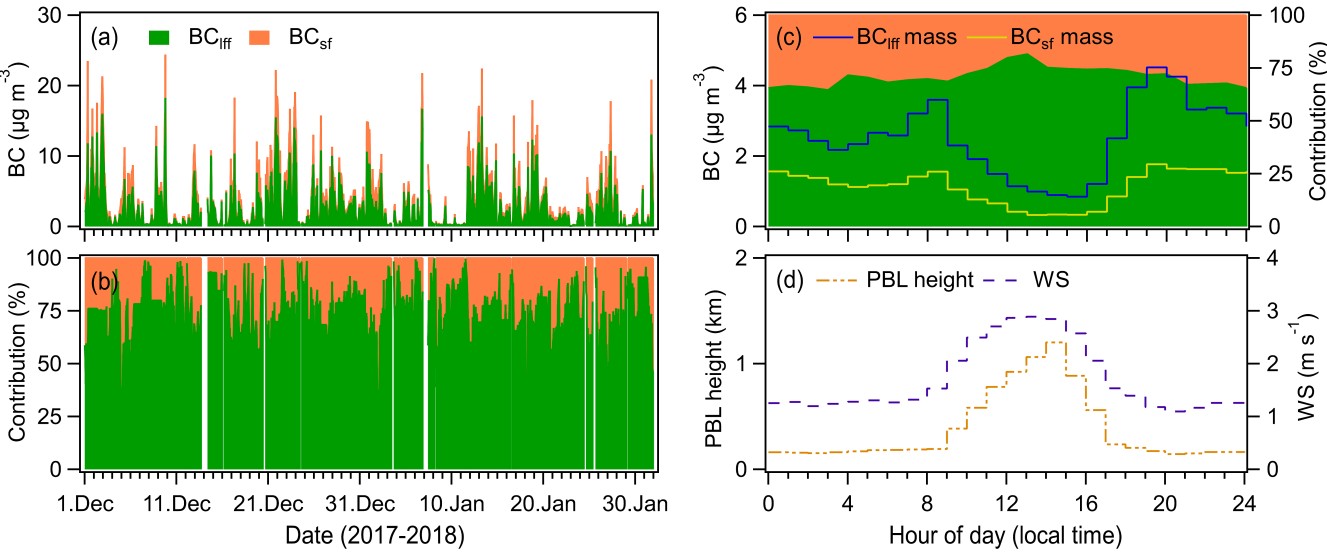

**Figure 2.** (left panel) (a) Time series of hourly averaged mass concentrations of black carbon (BC) aerosol from liquid fossil fuel ($BC_{lff}$) and solid fuel sources ($BC_{sf}$) and (b) their contributions to BC loading during the entire campaign. (right panel) Diurnal variations of (c) mass concentrations and mass fractions of $BC_{lff}$ and $BC_{sf}$ as well as (d) the planetary boundary layer (PBL) height and wind speed (WS) during the campaign.

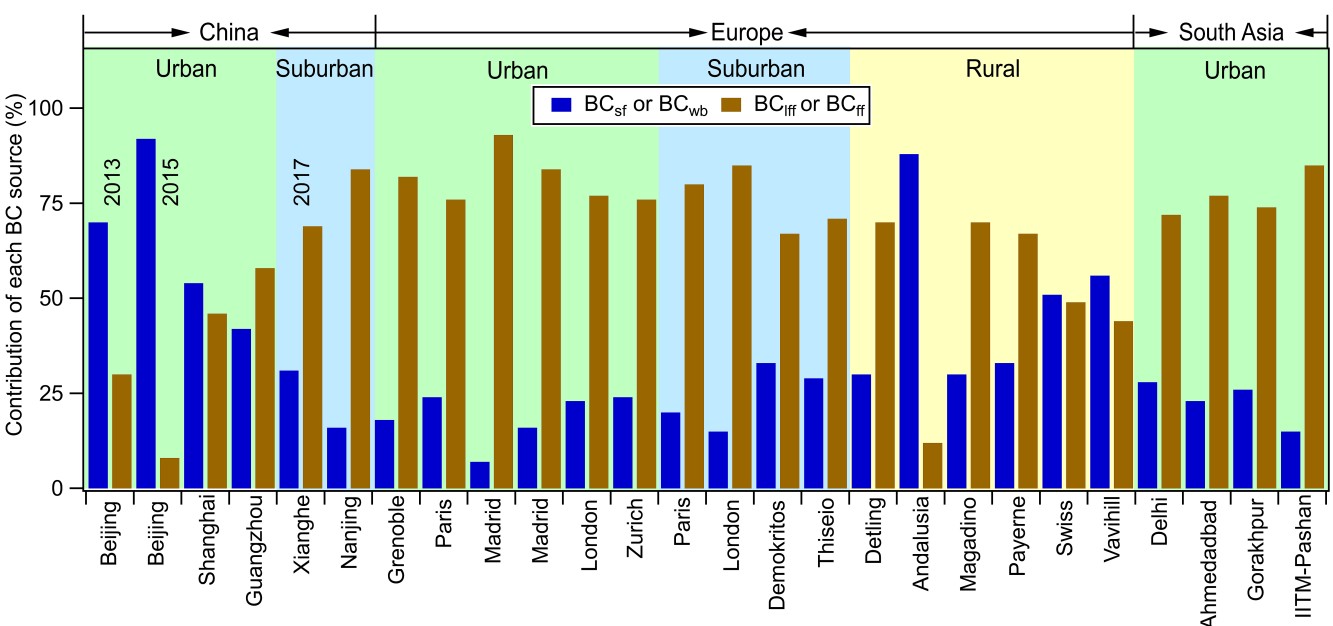

**Figure 3.** Comparison of different sources of black carbon (BC) in urban, suburban, and rural areas in China and Europe. $BC_{sf}$ and $BC_{wb}$ describe BC from solid fuel source and wood burning, respectively. $BC_{lff}$ and $BC_{ff}$ represent BC from liquid fossil fuel and fossil fuel sources, respectively. Detailed information of the data is summarized in Table S1.





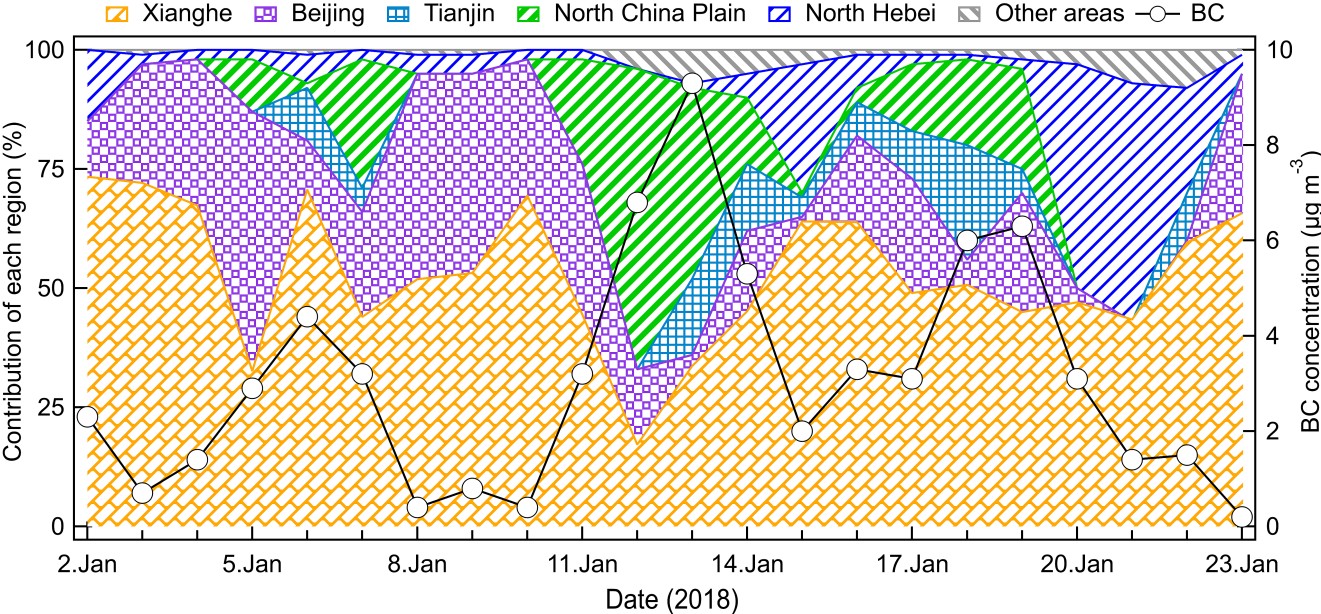

**Figure 4.** Time series of the mass concentration of black carbon (BC) and the contributions of each source region to BC loading during 2 – 23 January 2018.





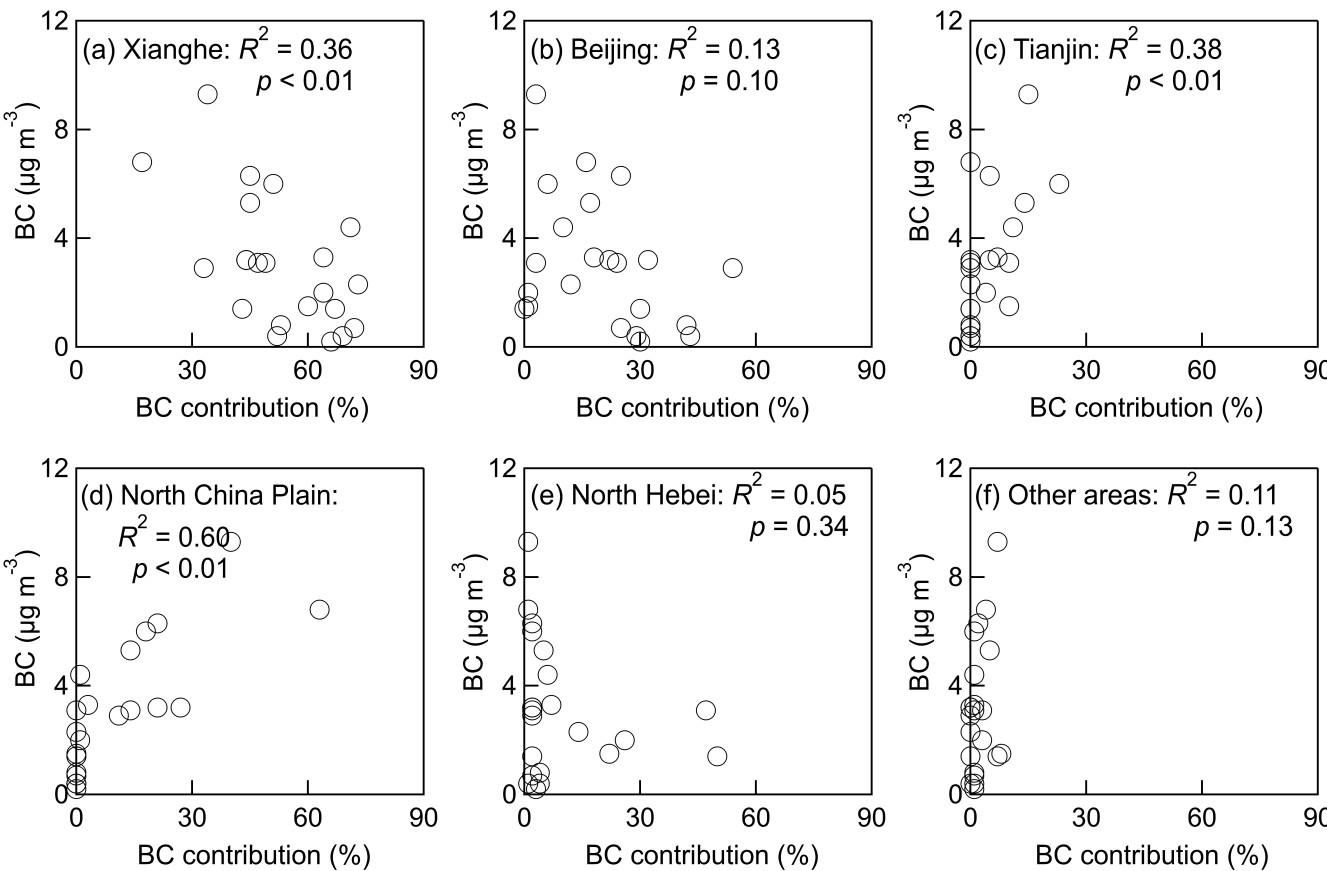

**Figure 5.** Scatter plots of the mass concentration of black carbon (BC) versus the BC contributions from different regions.





**Figure 6.** Daily average black carbon (BC) mass concentrations (μg m⁻³, represented by the color bar) simulated at Xianghe and surrounding areas from (a–d) 11 to 14 January 2018. The Weather Research and Forecasting model coupled to a chemistry (WRF-Chem) model was used for the simulation. The red rectangles represent different source regions, which is summarized in Table S2.





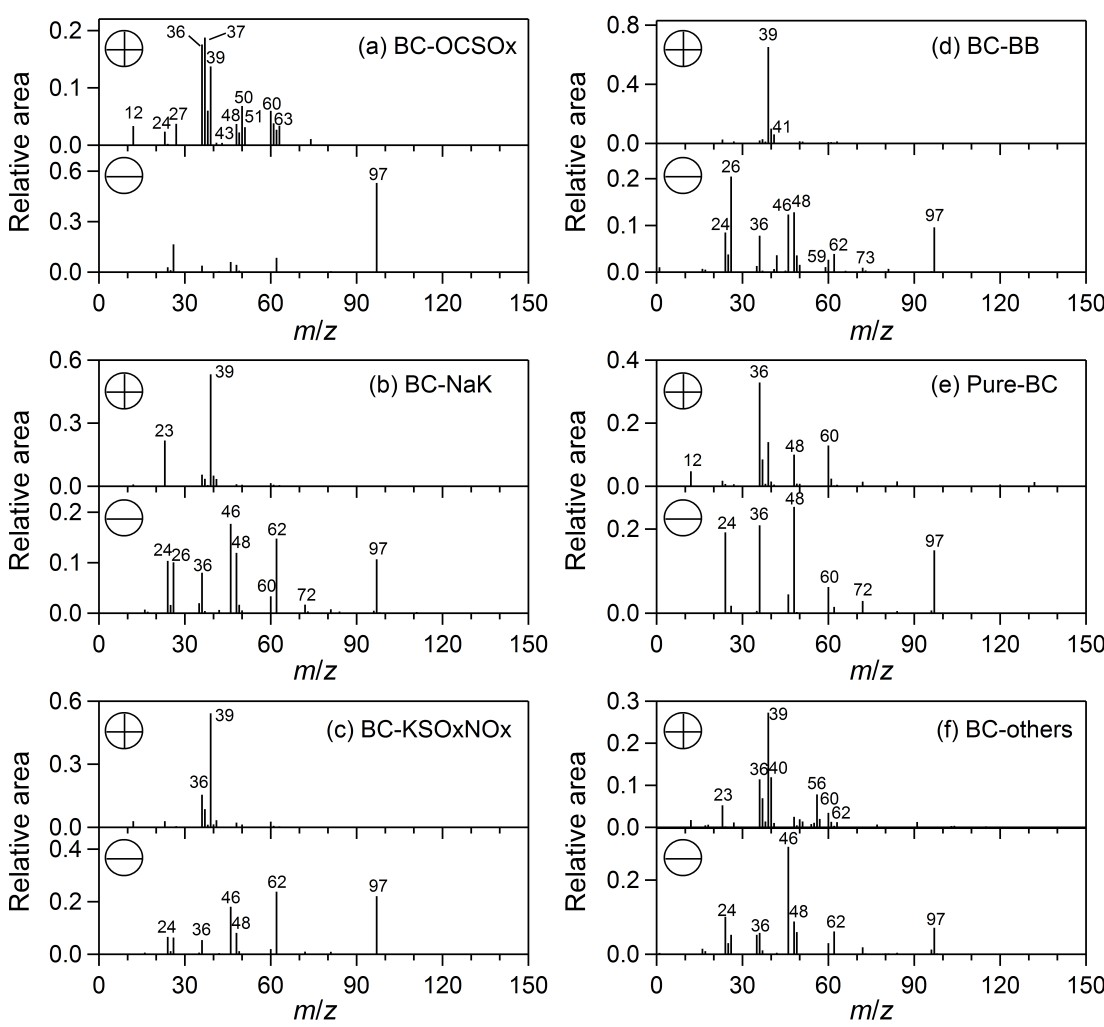

**Figure 7.** Average mass spectral pattern of six types of black carbon (BC)-containing particles.



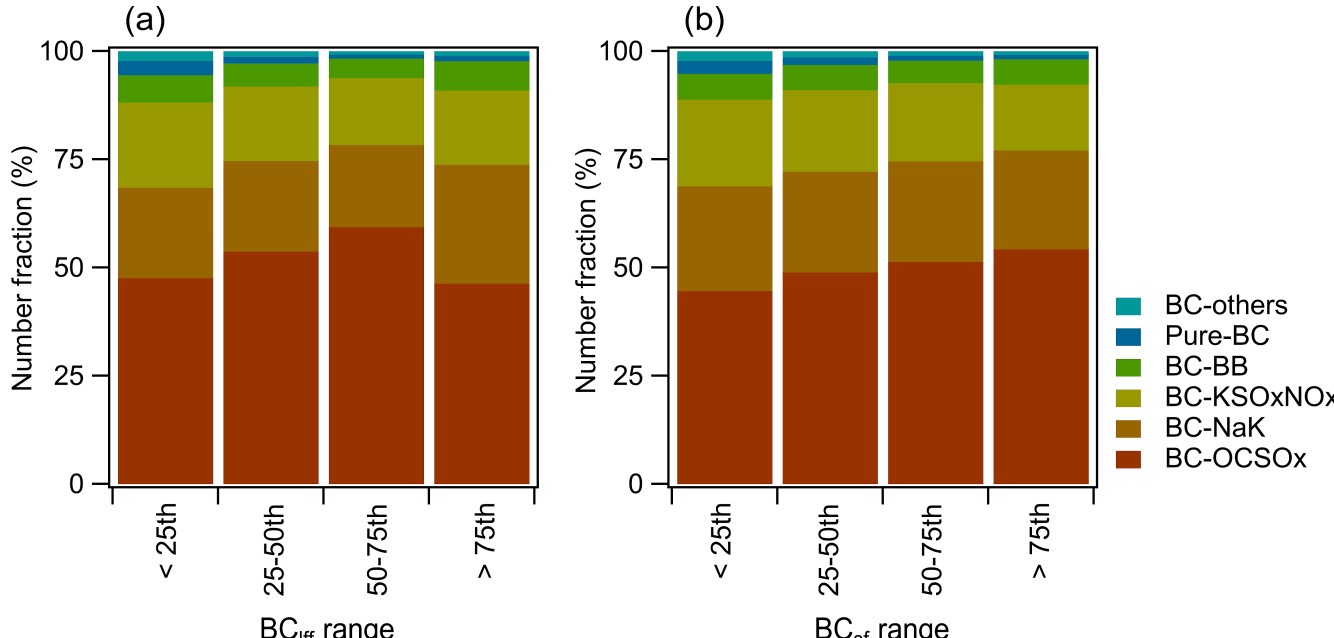

**Figure 8.** Number fractions of different BC-containing particle classes at different ranges of black carbon (BC) from (a) liquid fossil fuel ($BC_{lff}$) and (b) solid fuel sources ($BC_{sf}$). The 25th, 50th, and 75th denote the 25%, 50%, and 75% percentiles, respectively.

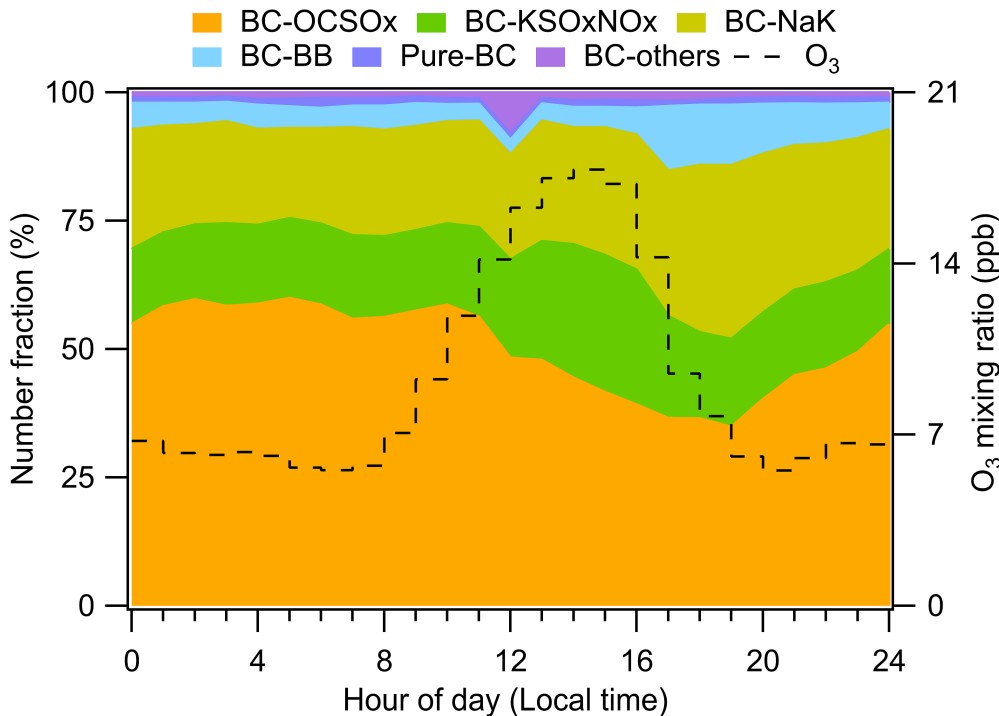

**Figure 9.** Diurnal patterns of total black carbon (BC)-containing particle count, number fraction of six types of BC-containing particles, and ozone ($O_3$) mixing ratio.



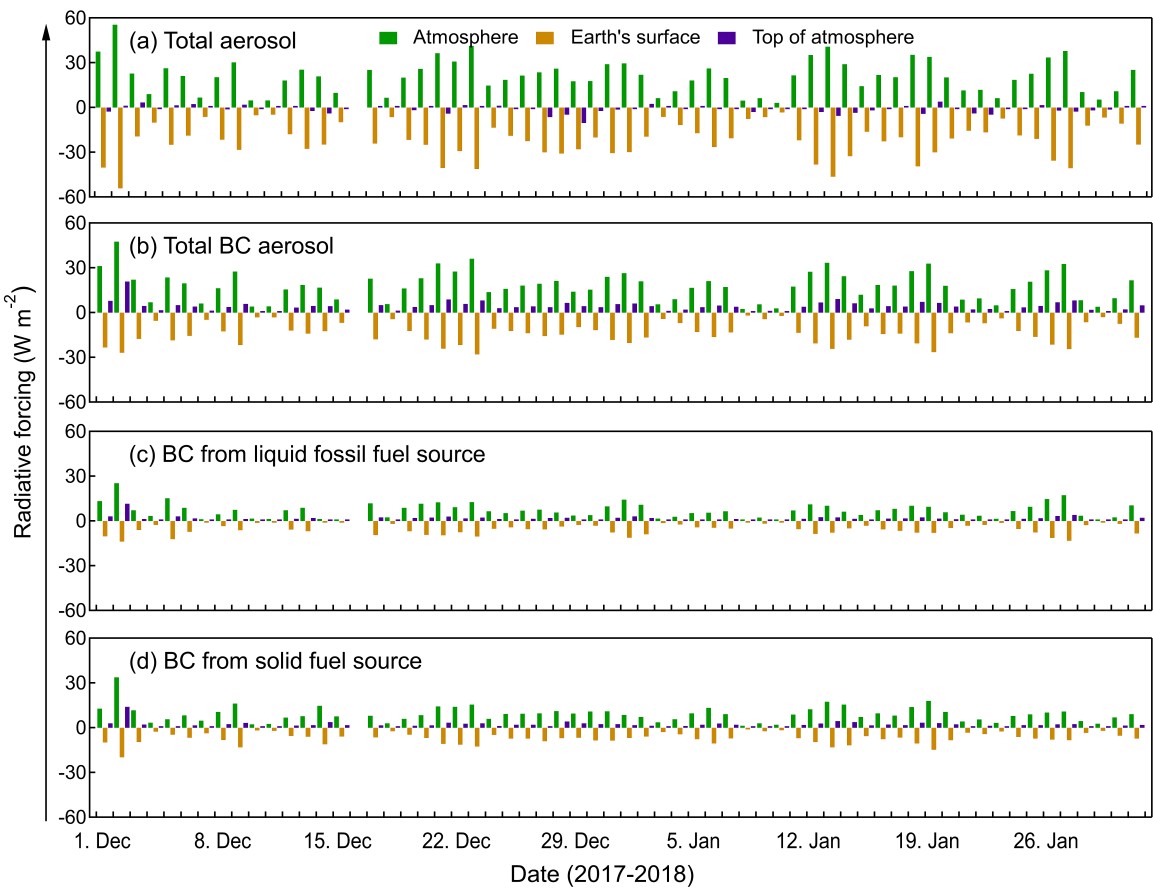

**Figure 10.** Time series of hourly averaged direct radiative forcing caused by (a) total aerosol and (b) black carbon (BC) as well as BC from (c) liquid fossil fuel and (d) solid fuel sources at the Earth's surface, the top of the atmosphere, and in the atmosphere.