# Peer review of "Measurement report: Source and mixing state of black carbon aerosol in the North China Plain: Implications for radiative effect"

_Atmospheric Chemistry and Physics, 2020_

## Referee Comment (RC1) · Anonymous Referee #1 · 12 Aug 2020

The manuscript by Wang et al. focused on the source apportionment and mixing state of black carbon in the North China Plain. The objectives of the study were to (a) determine the contributions of different sources and regions to the BC mass, (b) identify the chemical composition of BC coatings, and (c) evaluate the impacts of BC on regional radiative forcing. The authors used various methods to achieve these objectives. The major methods include a multi-wavelength optical approach combined with the source-based absorption Ångström exponent, WRF-Chem model, ART-2a, and SBDART model. The authors found that (a) the contribution of traffic emission was

dominant to BC mass in the study region; (b) BC-containing particles existed in six classes to the mixing state of organic and inorganic substances; (c) the estimated BC forcing was positive with +18.0 W m-2 and a heating rate of 0.5 K day-1 in the study region. The designed experiments were comprehensive, and the results were robust. The new data generated from this study should be valuable to understand the present status of the regional air quality and radiative forcing affected by BC in the North China Plain under the background of emission reduction. Therefore, I would suggest the manuscript for publication after the authors address the following specific comments.

(1) Although a photoacoustic instrument was used to correct the impact of filter matrix scattering of AE33 aethalometer, the PAX was operated at a single wavelength of 532 nm, which may bring uncertainty for other wavelengths of AE33 absorption. This should be pointed out in the study.

(2) A Nafion dryer was used for ACSM. How about the AE33, PAX, and SPAMS? High ambient humidity can also influence the measurements of these instruments. A schematic of the measurement system should be used to show the setup of each instrument.

(3) The authors should provide more information on bench tests, as the vehicle emission is a crucial component of the BC sources; relevant references should be cited.

(4) Insufficient detail is given regarding the radiative forcing calculations. The radiative forcing is related to the vertical information of the atmosphere. The authors used the OPAC model to retrieve the vertical optical parameters used in the SBDART model, but the brief description makes the calculation unclear. More information about the OPAC model should be added in the supporting information

(5) One finding of this study is the change of BC source in winter NCP through comparison with previous studies in this region. The authors considered this as a successful example of coal-to-gas switching to reduce pollutants under the new regulations by the Chinese State Council released in 2013. It would be useful to see a thorough comparison between the results of this study and those conducted before 2013 in this region. There should be relevant data from previous measurements by others in this region.

(6) The term 'Aethalometer model' gets misused in places. This refers to the data analysis technique, not the instrument itself.

(7) The map of Fig. S6 does not make much sense to someone unfamiliar with Chinese geography. Suggest showing a zoomed-out version showing the wider region.

(8). The authors may improve the manuscript content by avoiding grammar and spelling errors throughout the text. I'm not going to list them, but the manuscript should be checked carefully and polished by a native English speaker.

---

## Referee Comment (RC2) · Anonymous Referee #2 · 24 Aug 2020

Summary:

This paper discusses measurements of black carbon (BC) made with an AE33 aethalometer and a SP-AMS in the North China Plain (NCP) during winter in 2017-2018. These observations are discussed in the context of recent emission reduction policies. Initial laboratory experiments were made of different fuel sources in an environmental chamber to develop a model to interpret the aethalometer measurements. This model was then applied to 2 months of measurements at a ground site, and the results are interpreted in the context of WRF-Chem modeling results to estimate relative

contributions from regional sources and the BC radiative effect. The relative contributions of liquid fossil fuel sources and solid fuel sources were evaluated, and these relative sources were discussed in the context of their contribution to the regional BC radiative effect.

The observations and the analysis presented here are a valuable contribution to the literature, given the importance of BC as a climate forcer and the NCP as a significant anthropogenic source region for these aerosols. The conclusion that focusing on a reduction of BC from solid fuel sources could lead to greater gains than liquid fuels is an important conclusion for policy makers.

While it's clear that a significant amount of work went into the acquisition of measurements and the analysis of results presented in this study, the discussion is often challenging to follow. While each separate part of the measurements, analysis, and modeling were described in Section 2, it is often not clear which measurement or model contributed to the results discussed in Section 3. The paper would also benefit from a more clear discussion of the BC observations in light of best practices for discussing observations from different instruments established in the recent literature. In addition, the radiative forcing calculation is not clearly described.

General Comments:

Observations of BC with the different instruments used in the study should be more clearly distinguished (eBC, rBC) as discussed in Lack et al. 2014. This is an important issue when comparing observations between different instruments, as BC is operationally defined, and this point should be made clearly in the discussion. This is particularly confusing in the discussion of the source-specific AAE model, as it seems both observations from the aethalometer and the SP-AMS are used to determine the source-specific AAE values. It is also important in the context of the BC radiative effect calculation, as observations of BC mass loadings can differ by a factor of 2 or more when comparing different measurement techniques.

The discussion of the radiative forcing is not very clear, and I had a hard time following how this calculation was performed. More details need to be given. In addition, this appears to be a calculation of the BC radiative effect (rather than radiative forcing), as described in Heald et al. 2014.

Overall, the paper could benefit from English language editing.

Specific Comments:

The title is quite long.

Page 1, lines 23-25. It is unclear here how the "local emissions" differ from the "emissions in the NCP". If this is meant to differentiate local emissions relative to regional emissions this could be more clearly stated.

A schematic of the instrument setup during the environmental chamber experiments and the measurement campaign could be a helpful addition. Additionally the map showing the location of the measurement site (S1) would be helpful in the main text.

Some more details would be useful for the observations from the source emission experiments in order that these could be more easily compared with other similar experiments. At what point during the burns was the emitted smoke measured? Do the values in Table 1 represent average values of the aerosol optical properties during the entire period of the burn? What does the "test number" in Table 1 refer to? Is this the number of experiments performed? This should be made clear in the caption.

I found the discussion in 3.1.1 and Figure 1 to be unclear. Given the range of values for AAE for the different fuel types shown in Table 1, the limitations of the aethalometer model, which assumes a single value for AAE for liquid fuels and solid fuels, should be more clearly discussed in the text.

Does Table 2 give observations from the SP-AMS? As this discusses the number of particles, that would seem to be the case but this should be clearly stated (e.g. by referring to rBC rather than BC).

Page 11, lines 13. How was this percent contribution determined? Is this from the WRF-Chem model or does this use the observed BC? Is the BC concentration shown in Figure 5 from the model results or from the observations?

Figure 6. It would be useful to replace the Region 1-6 labels shown in the figure with the names of the regions used in the text (or include this information in the caption).

Figure 8. It is unclear what the percentile range on the x-axis refers to. Is this with respect to the BC mass loading?

It would be useful to discuss the results in Section 3.3 in the context of the different BC source emissions discussed in Section 3.1

Are the results in Section 3.4 from the WRF-Chem model? It is not clearly stated in the text. Also it would be useful to provide some context for this estimation of the BC radiative effect in terms of previous calculations in the literature for the NCP region.

References:

Heald, C.L., Ridley, D.A., Kroll, J.H., Barrett, S.R.H., Cady-Pereira, K.E., Alvarado, M.J. and Holmes, C.D., 2014. Contrasting the direct radiative effect and direct radiative forcing of aerosols.

Lack, D.A., Moosmüller, H., McMeeking, G.R., Chakrabarty, R.K. and Baumgardner, D., 2014. Characterizing elemental, equivalent black, and refractory black carbon aerosol particles: a review of techniques, their limitations and uncertainties. Analytical and bioanalytical chemistry, 406(1), pp.99-122.
* * *

---

## Author Response (AR1)

*The manuscript by Wang et al. focused on the source apportionment and mixing state of black carbon in the North China Plain. The objectives of the study were to (a) determine the contributions of different sources and regions to the BC mass, (b) identify the chemical composition of BC coatings, and (c) evaluate the impacts of BC on regional radiative forcing. The authors used various methods to achieve these objectives. The major methods include a multi-wavelength optical approach combined with the source-based absorption Ångström exponent, WRF-Chem model, ART-2a, and SBDART model. The authors found that (a) the contribution of traffic emission was dominant to BC mass in the study region; (b) BC-containing particles existed in six classes to the mixing state of organic and inorganic substances; (c) the estimated BC forcing was positive with $+18.0$ W m$^{-2}$ and a heating rate of 0.5 K day$^{-1}$ in the study region. The designed experiments were comprehensive, and the results were robust. The new data generated from this study should be valuable to understand the present status of the regional air quality and radiative forcing affected by BC in the North China Plain under the background of emission reduction. Therefore, I would suggest the manuscript for publication after the authors address the following specific comments.*

**Response:** The authors thank the reviewer's valuable suggestions, and we believe that the revised manuscript has been significantly improved after considering the comments. Below are the point-to-point responses, and the modifications to the manuscript are marked in the revised manuscript.

*(1) Although a photoacoustic instrument was used to correct the impact of filter matrix scattering of AE33 aethalometer, the PAX was operated at a single wavelength of 532 nm, which may bring uncertainty for other wavelengths of AE33 absorption. This should be pointed out in the study.*

**Response:** We agree with the reviewer. In the revised manuscript, we added a sentence to address the this uncertainty. It now reads as follows:

"As shown in Fig. S2, a 520 nm wavelength of AE33 absorption was strongly

correlated with the PAX absorption ($R^2 = 0.97$, $p < 0.01$). The slope of 2.57 was then used to correct the AE33 data. However, a single-wavelength-based correction method may result in underestimation at $\lambda = 370$ and $470$ nm, and overestimation at $\lambda = 590$, $660$, and $880$ nm (Kim et al., 2019)."

*(2) A Nafion dryer was used for ACSM. How about the AE33, PAX, and SPAMS? High ambient humidity can also influence the measurements of these instruments. A schematic of the measurement system should be used to show the setup of each instrument.*

**Response:** The collected ambient aerosols were dried before measurements by all instruments. We followed the reviewer's suggestion and added a schematic of the instrumental setups. Please see Figure R1 below (also see Figure S1 in the revised supplemental material).

[Figure]

**Figure R1.** Schematic presentation of the instrumental setups of the ambient aerosol measurements.

*(3) The authors should provide more information on bench tests, as the vehicle emission is a crucial component of the BC sources; relevant references should be cited.*

**Response:** Following the reviewer's suggestion, we added more description about the bench tests in the revised manuscript. It now reads as follows:

"The motor vehicle exhaust emissions were performed using a LDWJ6/135 detection system of loading and speed reduction on the light duty diesel vehicle (Shenzhen Huiyin Industrial Development Co., Ltd, Shenzhen, China). This system contains two different sizes of expansion cylinders that are used to carry the driving wheels of the vehicles. Fig. S5 shows the schematic presentation of the instrumental setup of motor vehicle exhaust emissions. Gasoline and diesel cars at idle and at different driving speeds (i.e., 20 and 40 km h$^{-1}$) were tested. The automobile exhaust smoke particles were collected using a particle sampling probe in the exhaust pipe. The particles were dried by a silica gel dryer before AE33 aethalometer measurement. The measured $b_{abs}(\lambda)$ used to estimate the AAE was averaged over the period that the driving speed was relatively stable."

*(4) Insufficient detail is given regarding the radiative forcing calculations. The radiative forcing is related to the vertical information of the atmosphere. The authors used the OPAC model to retrieve the vertical optical parameters used in the SBDART model, but the brief description makes the calculation unclear. More information about the OPAC model should be added in the supporting information.*

**Response:** Following the reviewer's suggestion, we added more description of the OPAC model in the revised supplemental material (Text S3). It reads as follows:

"The aerosol optical depth (AOD), single scattering albedo (SSA), and asymmetric parameter (AP) are important parameters used in the SBDART model to estimate aerosol radiative effect. In this study, these optical parameters were derived by the OPAC model. A detailed description of the software package of OPAC has been documented by Hess et al. (1998). The measured mass concentrations of OC, EC, and water-soluble ions as well as the estimated mineral dust (= [Fe]/0.035) were input in the OPAC model to estimate the AOD, SSA, and AP. The measurements of these chemical species are described in Text

S1. The number concentration of BC in the OPAC model was constrained by the measured EC mass concentration. Although several water-soluble ions and mineral dust were obtained, they did not cover all water-soluble and insoluble materials. Therefore, the number concentrations of water-soluble and insoluble materials were tuned for OPAC model based on the measured data. This was done by comparing the OPAC-derived light scattering, light absorption, and SSA with the corresponding PAX-measured ones until the differences were within 5% for each parameter. After the aerosol light extinction coefficient (sum of light scattering and absorption) was obtained, the AOD was then estimated as follows (Hess et al., 1998):

$$\text{AOD} = \sum_j \int_{H_{j,min}}^{H_{j,max}} \sigma_{e,j}(h)dh = \sum_j \sigma_{e,j}^1 N_j(0) \int_{H_{j,min}}^{H_{j,max}} e^{-\frac{h}{Z_j}} dh \qquad (S1)$$

where $H_{j,max}$ and $H_{j,min}$ were the upper and lower boundary in layer $j$; $\sigma_{e,j}$ was the surface aerosol light extinction coefficient in layer $j$; $h$ was the layer height; $\sigma_{e,j}^1$ represented the aerosol light extinction coefficient that was normalized to 1 particle cm$^{-3}$; $N_j$ was the number concentration in layer $j$; and $Z$ was the scale height. The OPAC-derived AODs were tuned to match the satellite-derived AODs (https://giovanni.gsfc.nasa.gov/giovanni) by altering the scale height in OPAC until the difference between them was within 5%. Owing to the closure with AOD and anchoring of the chemical composition, the assumptions in the OPAC model did not exhibit a significant impact on the estimation of radiative effect using SBDART model (Satheesh and Srinivasan 2006)."

*(5) One finding of this study is the change of BC source in winter NCP through comparison with previous studies in this region. The authors considered this as a successful example of coal-to-gas switching to reduce pollutants under the new regulations by the Chinese State Council released in 2013. It would be useful to see a thorough comparison between the results of this study and those conducted before 2013 in this region. There should be relevant data from previous measurements by*

*others in this region.*

**Response:** Following the reviewer's suggestion, we added BC source apportionment studies conducted before 2013 in the NCP region. This information was updated in Figure 4 in the revised manuscript (also see Figure R2 below).

[Figure]

**Figure R2.** Comparisons of the different sources of black carbon (BC) in urban, suburban, and rural areas in China and Europe. $BC_{sf}$ and $BC_{wb}$ describe BC from solid fuel sources and wood burning, respectively. $BC_{lff}$ and $BC_{ff}$ represent BC from liquid fossil fuel and solid fossil fuel sources, respectively. Detailed information of the data is summarized in Table S1.

*(6) The term 'Aethalometer model' gets misused in places. This refers to the data analysis technique, not the instrument itself.*

**Response:** To make it clearer, we changed the "Aethalometer model" to "multi-wavelength optical method" in the revised manuscript.

*(7) The map of Fig. S6 does not make much sense to someone unfamiliar with Chinese geography. Suggest showing a zoomed-out version showing the wider region.*

**Response:** Following the reviewer's suggestion, we modified the map of Fig. S6 in the revised manuscript. Please see Figure R3 below (also see Figure S8 in the revised supplemental material).

[Figure]

**Figure R3.** Division of different regions in the Weather Research and Forecasting model coupled with chemistry (WRF-Chem).

*(8). The authors may improve the manuscript content by avoiding grammar and spelling errors throughout the text. I'm not going to list them, but the manuscript should be checked carefully and polished by a native English speaker.*

**Response:** The revised manuscript was polished by an English language editing agency.

**Anonymous Referee #2**

*Summary:*

*This paper discusses measurements of black carbon (BC) made with an AE33 aethalometer and a SP-AMS in the North China Plain (NCP) during winter in 2017-2018. These observations are discussed in the context of recent emission reduction policies. Initial laboratory experiments were made of different fuel sources in an environmental chamber to develop a model to interpret the aethalometer measurements. This model was then applied to 2 months of measurements at a ground site, and the results are interpreted in the context of WRF-Chem modeling results to estimate relative contributions from regional sources and the BC radiative effect. The relative contributions of liquid fossil fuel sources and solid fuel sources were evaluated, and these relative sources were discussed in the context of their contribution to the regional BC radiative effect.*

*The observations and the analysis presented here are a valuable contribution to the literature, given the importance of BC as a climate forcer and the NCP as a significant anthropogenic source region for these aerosols. The conclusion that focusing on a reduction of BC from solid fuel sources could lead to greater gains than liquid fuels is an important conclusion for policy makers.*

*While it's clear that a significant amount of work went into the acquisition of measurements and the analysis of results presented in this study, the discussion is often challenging to follow. While each separate part of the measurements, analysis, and modeling were described in Section 2, it is often not clear which measurement or model contributed to the results discussed in Section 3. The paper would also benefit from a more clear discussion of the BC observations in light of best practices for discussing observations from different instruments established in the recent literature. In addition, the radiative forcing calculation is not clearly described.*

**Response:** The authors appreciate the reviewer's valuable suggestions, and we believe that the revised manuscript has been significantly improved after addressing the comments. Below are the point-to-point responses, and the modifications to the

manuscript are marked.

*General Comments:*

*(1) Observations of BC with the different instruments used in the study should be more clearly distinguished (eBC, rBC) as discussed in Lack et al. 2014. This is an important issue when comparing observations between different instruments, as BC is operationally defined, and this point should be made clearly in the discussion. This is particularly confusing in the discussion of the source-specific AAE model, as it seems both observations from the aethalometer and the SP-AMS are used to determine the source-specific AAE values.*

**Response:** We followed the reviewer's suggestion and clarified the term of BC measured by different instruments. According to the review studies of BC by Lack et al. (2014) and Petzold et al. (2013), we used "equivalent BC (eBC)" for AE33 aethalometer measurements and "elemental carbon (EC)-containing particles" for SPAMS measurements. The source-specific AAE was calculated based on the aerosol light absorption which was measured by AE33 aethalometer. We have clarified this in the revised manuscript. It now reads as follows:

> "Table 1 summarizes the average AAEs obtained from the sources of liquid fossil fuels and solid fuels. These source-specific AAEs were calculated using $b_{abs}(370)$ and $b_{abs}(880)$ (Eqs. 3 and 4)."

*(2) It is also important in the context of the BC radiative effect calculation, as observations of BC mass loadings can differ by a factor of 2 or more when comparing different measurement techniques.*

**Response:** We agree with the reviewer that different BC instruments induce uncertainties in BC measurements in studies. In the revised manuscript, we clarified this point, and it reads as follows:

> "Compared to previous DRE obtained from the SBDART model, the atmospheric DRE derived by eBC values in this study ($+18.0 \pm 9.6$ W m$^{-2}$) was

comparable to that of South China (+17.0 W m$^{-2}$, Huang et al., 2011) but was lower than that of Northwest China (+16.6 to +108.8 W m$^{-2}$, Zhao et al., 2019). In addition to the varying BC burden in different areas, the BC measurement techniques used in different studies may also contribute to the differences in BC DRE calculations."

*(3) The discussion of the radiative forcing is not very clear, and I had a hard time following how this calculation was performed. More details need to be given. In addition, this appears to be a calculation of the BC radiative effect (rather than radiative forcing), as described in Heald et al. 2014.*

**Response:** After carefully read the study of Heald et al. (2014), we changed the original used 'radiative forcing' to 'radiative effect' in the revised manuscript. Additionally, we added more detailed description about the calculation of BC radiative effect. It now reads as follows:

"Aerosol direct radiative effect (DRE) (Heald et al. 2014) at the top of the atmosphere (TOA) or at the Earth's surface (ES) is the difference between the incoming (↓) and outgoing (↑) solar fluxes (F) with and without aerosols:

$$DRE = (F\downarrow - F\uparrow)_{\text{with aerosol}} - (F\downarrow - F\uparrow)_{\text{without aerosol}} \qquad (8)$$

The aerosol DRE in the atmosphere was calculated by subtracting the DRE at the Earths' surface from the DRE at the top of the atmosphere.

In this study, the Santa Barbara DISORT Atmospheric Radiative Transfer (SBDART) model that was developed by Ricchiazzi et al. (1998) was used to perform the radiative transfer calculations in the shortwave spectral region of 0.25–4.0 μm. The SBDART model is a widely used tool for estimating aerosol DRE in the atmosphere (e.g., Zhang et al., 2017; Rajesh and Ramachandran, 2018; Boiyo et al., 2019). A detailed description of this model can be found in Ricchiazzi et al. (1998). The aerosols' optical depth, single scattering albedo, and asymmetric parameters are essential input factors in the SBDART model. These

optical parameters were estimated using the Optical Properties of Aerosols and Clouds (OPAC) model (Hess et al., 1998). Detailed calculations are shown in Text S3. Moreover, the surface albedo, solar zenith angle, and atmospheric parameter profiles are also important input factors in the SBDART model. The surface albedo was derived from the Moderate Resolution Imaging Spectroradiometer (https://modis-atmos.gsfc.nasa.gov/ALBEDO/index.html, last access: November 2019). The solar zenith angle was estimated using the latitude, longitude, and sampling time of the location. The atmospheric vertical profiles (including vertical distributions of temperature, pressure, water vapor, and ozone density) of mid-latitude winter embedded in the SBDART model were used."

*(4) Overall, the paper could benefit from English language editing.*

**Response:** The revised manuscript was polished by an English language editing agency.

*Specific Comments:*
*(5) The title is quite long.*

**Response:** We shortened the title as follows:

"Measurement report: Source and mixing state of black carbon aerosol in the North China Plain: Implications for radiative effect"

*(6) Page 1, lines 23-25. It is unclear here how the "local emissions" differ from the "emissions in the NCP". If this is meant to differentiate local emissions relative to regional emissions this could be more clearly stated.*

**Response:** Yes, it means to differentiate local emissions from regional emissions. To make it clearer, we revised this sentence in the manuscript. It now reads as follows:

"The air quality model indicated that local emissions were the dominant contributors to eBC at the measurement site. However, regional emissions from NCP were a critical factor for high eBC pollution."

*(7) A schematic of the instrument setup during the environmental chamber experiments and the measurement campaign could be a helpful addition. Additionally the map showing the location of the measurement site (S1) would be helpful in the main text.*

**Response:** Following the reviewer's suggestion, we have added the schematic of instrumental setups of our ambient measurements (see Figure R1 below and Figure S1) and source experiments (see Figure R2 below and Figure S4) in the revised supplementary material. Moreover, map of the sampling location was put in the main text (see Figure 1 in the revised main text).

[Figure]

**Figure R1.** Schematic presentation of the instrumental setups of the ambient aerosol measurements.

[Figure]

**Figure R2.** Schematic presentation of the instrumental setups of source experiments of biomass burning and coal combustion.

*(8) Some more details would be useful for the observations from the source emission experiments in order that these could be more easily compared with other similar experiments. At what point during the burns was the emitted smoke measured?*

**Response:** We thank the reviewer's insightful suggestion. In the revised manuscript, we added more description of the source emission experiments, which includes the information concerned by the reviewer. It now reads as follows:

"A custom-made passivated aluminum chamber (~8 m$^3$) was used to characterize the emission of solid fuels (i.e., biomass and coal) (Fig. S4). Performance evaluation of this chamber was done by Tian et al. (2015). Several types of biomass residues (wheat straw, rice straw, and corn stalk, cotton stalk, sesame stalk, soybean straw, and firewood) and coal (bituminous coal and honeycomb briquet) were used to represent biomass burning and coal combustion that occurs in the NCP. Each weighted sample was burned on a platform or in a stove that was placed inside the combustion chamber. For

biomass burning, the chamber background $b_{abs}(\lambda)$ was measured by AE33 aethalometer before ignition. When the background $b_{abs}(\lambda)$ was close to zero and stable, a propane torch was used to ignite the biomass on the platform. For coal combustion, a burned-out honeycomb coal in the stove was used as the igniter after the background $b_{abs}(\lambda)$ was small and stable in the chamber. The emitted smokes of each burn test were first diluted by a Model 18 dilution sampler (Baldwin Environmental Inc., Reno, NV, USA) before AE33 aethalometer measurements (Fig. S4). The $b_{abs}(\lambda)$ used to estimate the AAE was averaged over the entire period of each burn from ignition to $b_{abs}(\lambda)$ back to the background.

The motor vehicle exhaust emissions were performed using a LDWJ6/135 detection system of loading and speed reduction on the light duty diesel vehicle (Shenzhen Huiyin Industrial Development Co., Ltd, Shenzhen, China). This system contains two different sizes of expansion cylinders that are used to carry the driving wheels of the vehicles. Fig. S5 shows the schematic presentation of the instrumental setup of motor vehicle exhaust emissions. Gasoline and diesel cars at idle and at different driving speeds (i.e., 20 and 40 km h$^{-1}$) were tested. The automobile exhaust smoke particles were collected using a particle sampling probe in the exhaust pipe. The particles were dried by a silica gel dryer before AE33 aethalometer measurement. The measured $b_{abs}(\lambda)$ used to estimate the AAE was averaged over the period that the driving speed was relatively stable."

*(9) Do the values in Table 1 represent average values of the aerosol optical properties during the entire period of the burn? What does the "test number" in Table 1 refer to? Is this the number of experiments performed? This should be made clear in the caption.*

**Response:** Yes, the aerosol optical properties in Table 1 were calculated using the average light absorption of the entire period of each burn. The test number denotes the number of performed experiments. We added notes in the revised Table 1 (also see Table R1 below).

**Table 1.** Summary of aerosol absorption Ångström exponent (AAE) obtained from source experiment.

| | Solid fuel | | | | Liquid fuel | |
|---|---|---|---|---|---|---|
| | Crop residues | Firewood | Bituminous coal | Honeycomb briquet | Gasoline | Diesel |
| Maximum | 3.3 | 3.2 | 1.4 | 5.2 | 1.5 | 1.3 |
| Minimum | 1.6 | 2.7 | 1.0 | 2.2 | 1.4 | 1.0 |
| Average[a] | 2.4 | 2.9 | 1.1 | 4.0 | 1.5 | 1.2 |
| S.D.[b] | 0.4 | 0.2 | 0.2 | 0.9 | 0.1 | 0.1 |
| Test number[c] | 30 | 4 | 4 | 16 | 3 | 7 |

[a]The average value was calculated using the light absorption of the entire period of each burn.
[b]S.D. represents standard deviation.
[c]Test number denotes the number of performed experiments.

*(10) I found the discussion in 3.1.1 and Figure 1 to be unclear. Given the range of values for AAE for the different fuel types shown in Table 1, the limitations of the aethalometer model, which assumes a single value for AAE for liquid fuels and solid fuels, should be more clearly discussed in the text.*

**Response:** From the computational equations, the limitation of 'aethalometer model' is mainly from the applied source-specific AAE. Due to the high variation in the solid fuel AAE (e.g., 1.1–4.0 in this study), the selection of different value in the 'aethalometer model' can cause uncertainties in estimation of contribution from each source (e.g., solid fuels and liquid fuels) to total BC mass. Most of the current studies used source-specific AAEs from previous publication (e.g., Healy et al., 2017; Rajesh and Ramachandran, 2018; Zheng et al., 2019). Some other studies obtained the source-specific AAE through comparison with results from external source apportionment methods, such as ACSM-based organic aerosol (OA) sources (Ealo et al., 2016) and $^{14}$C technique (Martinsson et al., 2017; Zotter et al., 2017). In this study, we conducted the source experiments to obtain the possible suitable source-specific AAEs. In addition, we performed sensitivity analyses by comparing results of BC source apportionment with different OA sources (i.e., HOA and BBOA+CCOA) to verify the rationality of the applied AAE values. In the revised manuscript, we added

the limitation of the 'aethalometer model'. And based on the comment from Reviewer 1, we changed the term "Aethalometer model" to "multi-wavelength optical method" in the revised manuscript. It now reads as follows:

"From Eqs. 1–4 of the multi-wavelength optical method, its limitation is attributed to the choice of source-specific AAE. Since AAE exhibited high variations (e.g., 1.1–4.0 in this study), different AAE selections may lead to uncertainties when estimating the contributions of solid fuels and liquid fuels to eBC mass. In this study, the obtained average $AAE_{lff}$ (1.3) and $AAE_{sf}$ (2.8) were applied in the multi-wavelength optical method to obtain eBC source apportionment. A sensitivity test for each eBC source and organic aerosol (OA) subtype was further performed to verify the rationality of the used AAEs."

*(11) Does Table 2 give observations from the SP-AMS? As this discusses the number of particles, that would seem to be the case but this should be clearly stated (e.g. by referring to rBC rather than BC).*

**Response:** Yes, Table 2 shows the observations from SPAMS. As replied in comment (1) above, the 'BC' in table was revised to 'EC' as shown in Table R2 (also see Table 2 in the revised manuscript).

**Table R2.** Summary of names, numbers, and fractions of six types of elemental carbon (EC)-containing particles determined by a single particle aerosol mass spectrometer.

| Group | Number of particles | Fraction of particles (%) |
|---|---|---|
| EC internally mixed with OC and sulphate (EC-OCSOx) | 235874 | 51.9 |
| EC internally mixed with Na and K (EC-NaK) | 107272 | 23.6 |
| EC internally mixed with K, sulphate, and nitrate (EC-KSOxNOx) | 75227 | 16.6 |
| EC from biomass burning (EC-BB) | 26307 | 5.8 |
| Pure-EC | 5083 | 1.1 |
| Unidentified EC (EC-others) | 4670 | 1 |
| Total EC-containing | 454433 | 100 |

*(12) Page 11, lines 13. How was this percent contribution determined? Is this from the WRF-Chem model or does this use the observed BC? Is the BC concentration shown in Figure 5 from the model results or from the observations?*

**Response:** The percent contributions of local emissions and regional transport were obtained from WRF-Chem model. The BC concentration shown in Figure 5 was the observation values. We clarified in the manuscript, and it now reads as follows:

"As shown in Fig. S8, six source regions were identified in the WRF-Chem model to quantify the contributions of local emissions and regional transport to observed eBC mass. The information on each source region is summarized in Table S2, and their contributions to observed eBC mass are shown in Fig. 5."

[Figure]

**Figure R3.** Scatter plots of the measured mass concentrations of equivalent black carbon (eBC) versus the eBC contributions of different source regions obtained by WRF-Chem model.

*(13) Figure 6. It would be useful to replace the Region 1-6 labels shown in the figure with the names of the regions used in the text (or include this information in the caption).*

**Response:** Following the reviewer's suggestion, we added this information in the figure caption. Please see the Figure R4 below (also see Figure 7 in the revised manuscript).

[Figure]

**Figure R4.** Distributions of the daily average mass concentrations of equivalent black carbon (eBC) (µg m⁻³, represented by the color bar) at Xianghe and surrounding areas from 11th to 14th January, 2018 simulated by WRF-Chem model. The arrow denotes the wind speed. The red rectangles represent different source regions, which Region 1 is Xianghe, Region 2 is Beijing, Region 3 is Tianjin, Region 4 is North China Plain, Region 5 is North Hebei Province, and Region 6 is other areas except Region 1–5.

*(14) Figure 8. It is unclear what the percentile range on the x-axis refers to. Is this with respect to the BC mass loading?*

**Response:** Yes, the percentile range is the BC mass loading of different sources. We revised this figure to make it clearer. Please see Figure R5 below (also see Figure 9 in the revised manuscript).

[Figure]

**Figure R5.** Number fractions of elemental carbon (EC)-containing particle classes at different loading ranges of equivalent black carbon (eBC) from sources of (a) liquid fossil fuels ($eBC_{lff}$) and (b) solid fuels ($BC_{sf}$). The 25th, 50th, and 75th denote the 25%, 50%, and 75% percentiles, respectively.

*(15) It would be useful to discuss the results in Section 3.3 in the context of the different BC source emissions discussed in Section 3.1*

**Response:** We thank the reviewer's suggestion. As shown in Figure R5 above, we actually discussed the characteristics of chemical composition of EC-containing particles in the context of the different eBC sources in the original manuscript. We revised the manuscript in this part to make the discussion clearer. It now reads as follows:

"Fig. 9 shows the number of fractions of each class of EC-containing particles at different ranges of $eBC_{lff}$ and $eBC_{sf}$. The EC-OCSOx number fraction increased as $eBC_{sf}$ increased. In contrast, it dropped when $eBC_{lff}$ was higher than the value of the 75th percentile of $BC_{lff}$. This indicated a greater impact of solid fuel source on EC-OCSOx at a high eBC loading environment compared to the liquid fossil fuel source."

"The number fraction of EC-NaK increased with an increase in $BC_{lff}$ but kept stable with $BC_{sf}$ as shown in Fig. 9. These results demonstrate that EC-NaK was

likely associated with fresh traffic emissions than from solid fuels."

*(16) Are the results in Section 3.4 from the WRF-Chem model? It is not clearly stated in the text.*

**Response:** The results of radiative effect in Section 3.4 were estimated from the SBDART model. We added a sentence at the beginning of this paragraph to clarify this in the revised manuscript:

"Fig. 11 shows the eBC DRE variations as estimated by the SBDART model."

*(17) Also it would be useful to provide some context for this estimation of the BC radiative effect in terms of previous calculations in the literature for the NCP region.*

**Response:** We followed the reviewer's suggestion and added some comparisons of BC radiative effect from previous studies. It now reads as follows:

"In contrast, eBC exhibited a DRE range of +0.6 to +20.8 W m$^{-2}$ with an average of +4.4 ± 3.0 W m$^{-2}$ at the TOA, indicating a net energy gain and warm effect. This was attributed to the strong BC light absorption property that can impede the back scattered radiation reaching the TOA. The eBC DRE at the TOA in this study was comparable to the value over the NCP region (+6 to +8 W m$^{-2}$, Li et al., 2016)."

"Compared to previous DRE obtained from the SBDART model, the atmospheric DRE derived by eBC values in this study (+18.0 ± 9.6 W m$^{-2}$) was comparable to that of South China (+17.0 W m$^{-2}$, Huang et al., 2011) but was lower than that of Northwest China (+16.6 to +108.8 W m$^{-2}$, Zhao et al., 2019)."

Correspondence to: Qiyuan Wang (wangqy@ieecas.cn) and Junji Cao (cao@loess.llqg.ac.cn)

**Abstract.** Establishment of the sources and mixing state of black carbon (BC) aerosol is essential for assessing its impact on air quality and climatic effects. A winter campaign (December 2017–January 2018) was performed in the North China Plain (NCP) to evaluate the sources, coating composition, and radiative effects of BC under the background of emission reduction. Results showed that the sources of liquid fossil fuels (i.e., traffic emissions) and solid fuels (i.e., biomass and coal burning) contributed 69% and 31% to the total equivalent BC (eBC) mass, respectively. These values were arrived at by using a combination of multi-wavelength optical approach with the source-based aerosol absorption Ångström exponent values. The air quality model indicated that local emissions were the dominant contributors to eBC at the measurement site. However, regional emissions from NCP were a critical factor for high eBC pollution. A single particle aerosol mass spectrometer identified six classes of elemental carbon (EC)-containing particles. They included: EC coated by organic carbon and sulphate (52% of total EC-containing particles), EC coated by Na and K (24%), EC coated by K, sulphate, and nitrate (17%), EC associated with biomass burning (6%), pure-EC (1%), and others (1%). Different BC sources exhibited distinct impacts on the EC-containing particles. A radiative transfer model showed that the amount of detected eBC can produce an atmospheric direct radiative effect of +18.0 W m$^{-2}$ and a heating rate of 0.5 K day$^{-1}$. This study shows

删除了: **Evaluation of s…ources and mixing state of black carbon aerosol under the background of emission reduction …n the North China Plain: Ii

删除了: Regional Climate-Environment Research for Temperate East Asia

删除了: County, Hebei Province,

删除了: Accurate understanding …f the sources and mixing state of black carbon (BC) aerosol is essential for assessing its impacts…on air quality and climatic effects. Here, a… winter campaign (December 2017 … …anuary 2018) was conducted …erformed in the North China Plain (NCP) to evaluate the sources, coating composition, and radiative effects of BC under the background of emission reduction since 2013… Results showed that the sources of liquid fossil fuel source… (i.e., traffic emissions) and solid fuel source… (i.e., biomass and coal burning) contributed 69% and 31% the total equivalent BC (eBC) mass, respectively. These values were arrived at by,…using a combination of multi-wavelength optical approach combined …ith the source-based aerosol absorption Ångström exponent values. The air quality model indicated…that local emissions was …ere the dominant contributors to eBC at the measurement site on average,… Hh…wever, regional emissions in …romthe…NCP exerted …ere a critical role …actor for high eBC episode…ollution. A single particle aerosol mass spectrometer identified sS…x classes of elemental carbon (EC)BC…containing particles were identified,… They included:ing…(1) …B… coated by organic carbon and sulphate (52% of total EB…-containing particles), (2) …B… coated by Na and K (24%), (3) …B… coated by K, sulphate, and nitrate (17%), (4) B…C associated with biomass burning (6%), (5) P…ure-EB… (1%), and (6) …thers (1%). Different BC sources exhibitedhad…distinct impacts on those …he EB…-containing particles. A radiative transfer model estimated …howed that the amount of detected eBC detected …an produce an atmospheric direct radiative effectforcing

that reductions of solid fuel combustion-related BC may be an effective way of mitigating regional warming in the NCP.

**1 Introduction**

In the few past decades, black carbon (BC) aerosol has attracted considerable attention due to its substantial effects on the climate and atmospheric environment (Bond et al., 2013). It has strong light-absorption abilities that lead to substantive climate changes in the global atmosphere (+1.1 W m$^{-2}$). It is considered to be the second largest anthropogenic warming agent after carbon dioxide (Bond et al., 2013). In addition, the high atmospheric BC loading inhibits the development of a planetary boundary layer and enhances haze pollution (Ding et al., 2016). Reducing atmospheric BC loading is regarded as a win-win intervention for mitigating climate change and improving air quality (Kopp and Mauzerall, 2010).

Different emission sources (e.g., fossil fuel and biomass burning) and complex physicochemical properties (e.g., morphology, size, and coating composition) have hindered the efforts to assess the climatic and environmental impacts of BC (Vignati et al., 2010). To determine BC sources, current methods rely on data obtained from offline filter-based or online spectroscopic techniques (Briggs and Long, 2016). Among them, the carbon isotope approach (Zhang et al., 2015) and multi-wavelength optical method (Zotter et al., 2017) are often used to determine BC sources. The carbon isotope approach can be used to obtain relatively accurate results of BC sources. However, the analysis is limited by time resolution of the filter samples. The multi-wavelength optical method utilizes online data and has the advantage of a superior time resolution when determining BC sources. The principle of the multi-wavelength optical method is based on Beer-Lambert's Law. It utilizes measured aerosol light absorption at different wavelengths (Sandradewi et al., 2008). However, due to the lack of source-specific aerosol absorption Ångström exponent (AAE), studies often cite AAEs from previous literatures with fuel types being distinct among studies (Kalogridis et al., 2018; Zheng et al., 2019). These components induce potential uncertainties in using the multi-wavelength optical method. Therefore, a diverse set of AAEs from different source emissions are needed to improve the performance of this method.

Black carbon mixing state is whereby another chemical composition is coated on BC particles (internally mixed) or exists as separate particles (externally mixed). Freshly emitted BC particles (e.g., diesel vehicle

删除了: Results presented herein highlight …hat further …eductions of solid fuel combustion-related BC may be an more …ffective way to …f mitigatinge…regional warming in the NCP, although larger BC contribution was from liquid fossil fuel source

删除了: Over …n the few past decades, black carbon (BC) aerosol has attracted considerable attention due to its substantial effects on the climate and atmospheric environment (Bond et al., 2013). It has BC can affect climate via direct or semidirect radiative effects or effects or indirect cloud-albedo effects (Boucher et al., 2013). Due to its …trong light-absorptionbing…abilities that lead toy, BC can produce a…substantiveal…climate forcing …hangesglobally…in the present-day…lobal atmosphere (+1.1 W m$^{-2}$). , and hence i…t is considered as …o be the second largest anthropogenic warming agent after carbon dioxide (Bond et al., 2013). Moreover…n addition, the high atmospheric BC loading can depress…nhibits the development of a planetary boundary layer and aggravate …nhances haze pollution (Ding et al., 2016). Due to the short atmospheric residence time, r…educing atmospheric BC loading is regarded as a win-win policy …ntervention to …or mitigatinge…climate change and improvinges…

删除了: Owing to BC's various…ifferent emission sources (e.g., fossil fuel and biomass burning) and complex physicochemical properties (e.g., morphology, size, and coating composition) have hindered the efforts to, large uncertainty still remains in…assess theing its…climatice…and environmental impacts of BC (Vignati et al., 2010). To determine For …C source apportionment…, current methods relyare usually based…on the…data obtained from offline filter-based or online spectroscopicy …techniques (Briggs and Long, 2016). Among them, the carbon isotope approach (e.g., Δ$^{14}$C…hang et al., 2015) and multi-wavelength optical method (e.g., aethalometer model…otter et al., 2017) are often used to quantify …etermine BCthe…sources of BC (e.g., Zotter et al., 2017; Zhang et al., 2015)… The carbon isotope approach could …an be used to obtain relatively accurate results of BC source apportionment…. However, the analysis is limited by time resolution of the filter samples. The multi-wavelength optical methodaethalometer model which…utilizes online data and,…has the advantage of a superior time resolution whenin…determining BC sources. The principle of the multi-wavelength optical methodaethalometer model…is based on the…Beer-Lambert's Law. It utilizes using…measured aerosol light absorption at different wavelengths (Sandradewi et al., 2008). However, due to the lack of source-specific aerosol absorption Ångström exponent (AAE), a number of …tudies often adopted …ite AAEs in aethalometer model cited …rom previous literatures even the…ith fuel types are …eing distinct among studies (e.g., …alogridis et al., 2018; Zheng et al., 2019). These componentsis may…induce a large…otential uncertaintiesy…in using the multi-wavelength optical method 
[revised manuscript text omitted]

删除了: several

删除了: Aethalometer model

删除了: An aethalometer model

删除了: applied

删除了: contributions

删除了: the model

删除了: As demonstrated previously,

删除了: can be contributed

删除了: Due to the small

删除了: (11% of PM$_{2.5}$ mass)

删除了: and its small mass absorption cross section

删除了: (

删除了: ,

删除了: ,

删除了: caused by

删除了: can be neglected

删除了: Previous studies have demonstrated that the $b_{abs}(880)$ is mainly contributed by BC aerosol, while $b_{abs}(370)$ is associated with BC, primary and secondary brown carbon

and primary brown carbon (BrC)) and secondary formation (i.e., secondary BrC) (Laskin et al., 2015). Therefore, the $b_{abs}(370)$ and $b_{abs}(880)$ could be calculated from the perspective of emission sources as follows:

$$b_{abs}(880)_{lff} + b_{abs}(880)_{sf} = b_{abs}(880) \tag{1}$$

[revised manuscript text omitted]

删除了: signal …n the negative mass spectrum. This group was …xhibited the largest …ighest contribution to the total EC-containing particlesor,…constituting …52%, of the total BC-containing particles (…able 2), indicating that EB… was mainly coated with OC and sulphate. The presences of $^{43}$C$_2$H$_3$O$^+$ (… a marker denoting the oxidized organics,…(Gunsch et al., 2018), and $^{97}$HSO$_4^-$ impliedy…that EB…-OCSOx was …nderwent a certain degree of atmospheric aging processes… Fig. 9 shows the number of fractions of each class of EC-containing particles at different ranges of eBC$_{lff}$ and eBC$_{sf}$. As shown in Fig. 8, t…he EB…-OCSOx number fraction increased as eBC$_{sf}$ increased. In contrast, it dropped when eBC$_{lff}$ was larger …igher than the value of the 75th percentile of eBC$_{lff}$. This indicated…a greater impact of solid fuel source on EB…-OCSOx at a high eBC loading environment compared with …o the liquid fossil fuel source. The diurnal variations in EB…-OCSOx number fraction exhibited an upward trend at night after 19:00 (Fig. 109….,…This which was…ay be attributed to the

删除了: B…-NaK particles presented exhibited …trong signals of $^{23}$Na$^+$ and $^{39}$K$^+$ in the positive mass spectrum and less intense signals of $^{26}$CN$^-$, $^{46}$NO$_2^-$, $^{62}$NO$_3^-$, and $^{97}$HSO$_4^-$ in the negative mass spectrum. This group …xhibited the second largest toaccounting for 24% of…total EB…-containing particles (24%, 2). Intense signals of BC …arbon fragment ions ($m/z$ 24, 36, 48, 60, and 72) were concentrated …bserved inon…the negative mass spectrum. This,…indicateding…that EB…-NaK particles were mainly …reshly emitted. Meanwhile…dditionally, relatively higher…arger signals was …ere found for nitrate than …ompared to sulphate in the negative mass spectrum. However,, although…their signals were low. This finding wasis…consistent with the motor vehicle emissions, which were shown to contains…substantial nitrogen oxides (May et al., 2014). Furthermore, t…he number fraction of EB…-NaK enhanced …ncreased with an increase inas…eBC$_{lff}$ increased …ut kept stable with eBC$_{sf}$ as shown in (…ig. 98)… These results demonstrate that EB…-NaK was more …ikely associated with the…fresh traffic emissions than from relative to the …olid fuels emission

删除了: B…-KSOxNOx particles exhibited had a …trong signal of …$^9$K$^+$ signal in the positive mass spectrum and intense signals of …$^6$NO$_2^-$, $^{62}$NO$_3^-$, and $^{97}$HSO$_4^-$ signals in the negative mass spectrum. This group comprised …ccounted for 17% of the total EB…-containing particles (Table 2). The high signal intensities of nitrate and sulphate indicated that EB…-KSOxNOx particles suffered …nderwent substantive atmospherical…aging processes in the atmosphere… As shown in Fig. 10, tT…e number fraction of the EB…-KSOxNOx class was the only class …ne that increased in the afternoon, … This findingwhich…was consistent with the increase of ozone (O$_3$, Fig. 9… as measured with …y an ultraviolet photometric Model 49$i$ O$_3$ analyzer (Thermo Fisher Scientific, San Jose, CA, USA). This indicateds…that BC

[revised manuscript text omitted]

---

## Author Response (AR2)

*(1) p.8 line 19. "Earths'" should be "Earth's"*

**Response:** Change made. It now reads as follows:

"The aerosol DRE in the atmosphere was calculated by subtracting the DRE at the Earth's surface from the DRE at the top of the atmosphere."

*(2) p.10 lines 22-24. This sentence is unclear— if I understand what the authors are trying to convey here, "were belonged to the upper limit" should be "were determined from the upper limit"*

*"the coefficients of determining eBClff versus HOA (R2 = 0.60) and eBCsf versus (BBOA + CCOA) (R2 = 0.66) were belonged to the upper limit of all the R2 values obtained from different ranges of AAElff and AAEsf (Fig. 2)."*

**Response:** Change made. It now reads as follows:

"For the $AAE_{lff}$ (1.3) and $AAE_{sf}$ (2.8) used in this study, the coefficients of determining $eBC_{lff}$ versus HOA ($R^2 = 0.60$) and $eBC_{sf}$ versus (BBOA + CCOA) ($R^2 = 0.66$) were determined from the upper limit of all the $R^2$ values obtained from different ranges of $AAE_{lff}$ and $AAE_{sf}$ (Fig. 2)."

*(3) Sect. 2.4.2 and Sect. 3.2.*
*The simulated mass concentration of BC modeled by WRF-Chem would be more appropriately referred to as "BC" rather than "eBC".*

**Response:** The relevant content was modified in the revised manuscript.

[revised manuscript text omitted]